# Stable Vectorization of Multiparameter Persistent Homology using Signed Barcodes as Measures

**David Loiseaux[1*], Luis Scoccola[2*], Mathieu Carrière[1], Magnus B. Botnan[3], Steve Oudot[4]**

[1]DataShape, Centre Inria d'Université Côte d'Azur    [2]Mathematics, University of Oxford
[3]Mathematics, Vrije Universiteit Amsterdam    [4]GeomeriX, Inria Saclay and École polytechnique

## Abstract

Persistent homology (PH) provides topological descriptors for geometric data, such as weighted graphs, which are interpretable, stable to perturbations, and invariant under, e.g., relabeling. Most applications of PH focus on the one-parameter case—where the descriptors summarize the changes in topology of data as it is filtered by a single quantity of interest—and there is now a wide array of methods enabling the use of one-parameter PH descriptors in data science, which rely on the stable vectorization of these descriptors as elements of a Hilbert space. Although the multiparameter PH (MPH) of data that is filtered by several quantities of interest encodes much richer information than its one-parameter counterpart, the scarceness of stability results for MPH descriptors has so far limited the available options for the stable vectorization of MPH. In this paper, we aim to bring together the best of both worlds by showing how the interpretation of signed barcodes—a recent family of MPH descriptors—as signed measures leads to natural extensions of vectorization strategies from one parameter to multiple parameters. The resulting feature vectors are easy to define and to compute, and provably stable. While, as a proof of concept, we focus on simple choices of signed barcodes and vectorizations, we already see notable performance improvements when comparing our feature vectors to state-of-the-art topology-based methods on various types of data.

## 1 Introduction

### 1.1 Context

Topological Data Analysis (TDA) [26] is a field of data science that provides descriptors for geometric data. These descriptors encode *topological structures* hidden in the data, as such they are complementary to more common descriptors. And since TDA methods usually require the sole knowledge of a metric or dissimilarity measure on the data, they are widely applicable. For these reasons, TDA has found successful applications in a wide range of domains, including, e.g., computer graphics [58], computational biology [62], or material sciences [63], to name a few.

The mathematical definition of TDA's topological descriptors relies on *persistent homology*, whose input is a simplicial complex (a special kind of hypergraph) filtered by an $\mathbb{R}^n$-valued function. The choice of simplicial complex and function is application-dependent, a common choice being the complete hypergraph on the data (or some sparse approximation) filtered by scale and/or by some density estimator. The sublevel-sets of the filter function form what is called a *filtration*, i.e., a family $\{S_x\}_{x \in \mathbb{R}^n}$ of simplicial complexes with the property that $S_x \subseteq S_{x'}$ whenever $x \leq x' \in \mathbb{R}^n$ (where

---

*Equal contribution.

by definition $x \leq x' \Leftrightarrow x_i \leq x'_i \ \forall 1 \leq i \leq n$). Applying standard simplicial homology [45] with coefficients in some fixed field $\Bbbk$ to $\{S_x\}_{x \in \mathbb{R}^n}$ yields what is called a *persistence module*, i.e., an $\mathbb{R}^n$-parametrized family of $\Bbbk$-vector spaces connected by $\Bbbk$-linear maps—formally, a *functor* from $\mathbb{R}^n$ to the category vec of $\Bbbk$-vector spaces. This module encodes algebraically the evolution of the topology through the filtration $\{S_x\}_{x \in \mathbb{R}^n}$, but in a way that is neither succinct nor amenable to interpretation.

Concise representations of persistence modules are well-developed in the *one-parameter* case where $n = 1$. Indeed, under mild assumptions, the modules are fully characterized by their *barcode* [33], which can be represented as a point measure on the extended Euclidean plane [36]. The classical interpretation of this measure is that its point masses—also referred to as *bars*—encode the appearance and disappearance times of the topological structures hidden in the data through the filtration. Various *stability theorems* [23, 29, 30, 68] ensure that this barcode representation is *stable* under perturbations of the input filtered simplicial complex, where barcodes are compared using optimal transport-type distances, often referred to as Wasserstein or bottleneck distances. In machine learning contexts, barcodes are turned into vectors in some Hilbert space, for which it is possible to rely on the vast literature in geometric measure and optimal transport theories. A variety of stable vectorizations of barcodes have be designed in this way—see [3] for a survey.

There are many applications of TDA where *multiparameter persistence modules* (that is, when $n \geq 2$) are more natural than, and lead to improved performance when compared to, one-parameter persistence modules. These include, for example, noisy point cloud data [77], where one parameter accounts for the geometry of the data and another filters the data by density, and multifiltered graphs [34], where different parameters account for different filtering functions.

In the multiparameter however, the concise and stable representation of persistence modules is known to be a substantially more involved problem [19]. The main stumbling block is that, due to some fundamental algebraic reasons, there is no hope for the existence of a concise descriptor like the barcode that can completely characterize multiparameter persistence modules. This is why research in the last decade has focused on proposing and studying incomplete descriptors—see, e.g., [10] for a recent survey. Among these, the *signed barcodes* stand out as natural extensions of the usual one-parameter barcode [4, 11, 12, 49], being also interpretable as point measures. The catch however is that some of their points may have negative weights, so the measures are *signed*. As a consequence, their analysis is more delicate than that of one-parameter barcodes, and it is only very recently that their optimal transport-type stability has started to be understood [12, 55], while there currently is still no available technique for turning them into vectors in some Hilbert space.

## 1.2 Contributions

We believe the time is ripe to promote the use of signed barcodes for feature generation in machine learning; for this we propose the following pipeline (illustrated in Fig. 1):

$$\left\{ \begin{array}{c} \text{geometric} \\ \text{datasets} \end{array} \right\} \xrightarrow{\text{filtration}} \left\{ \begin{array}{c} \text{multifiltered} \\ \text{simplicial} \\ \text{complexes} \end{array} \right\} \xrightarrow{\text{homology}} \left\{ \begin{array}{c} \text{multiparameter} \\ \text{persistence} \\ \text{modules} \end{array} \right\} \xrightarrow{\text{signed barcode descriptor}} \left\{ \begin{array}{c} \text{signed} \\ \text{barcodes} \end{array} \right\} \xrightarrow{\text{vectorization}} \begin{array}{c} \text{Hilbert} \\ \text{space} \end{array}$$

The choice of filtration being application-dependent, we will mostly follow the choices made in related work, for the sake of comparison—see also [10] for an overview of standard choices. As signed barcode descriptor, we will mainly use the *Hilbert decomposition signed measure* (Definition 4), and when the simplicial complex is not too large we will also consider the *Euler decomposition signed measure* (Definition 5). These two descriptors are arguably the simplest existing signed measure descriptors, so they will serve as a proof of concept for our pipeline. They also offer the advantage of being efficiently computable, with effective implementations already available [42, 48, 65, 71]. With these choices of signed measure descriptors, the only missing step in our pipeline is the last one—the vectorization. Here is the summary of our contributions, the details follow right after:

- We introduce two general vectorization techniques for signed barcodes (Definitions 6 and 7).

- We prove Lipschitz-continuity results (Theorems 1 to 3) that ensure the robustness of our entire feature generation pipeline.

- We illustrate the practical performance of our pipeline compared to other baselines in various supervised and unsupervised learning tasks.

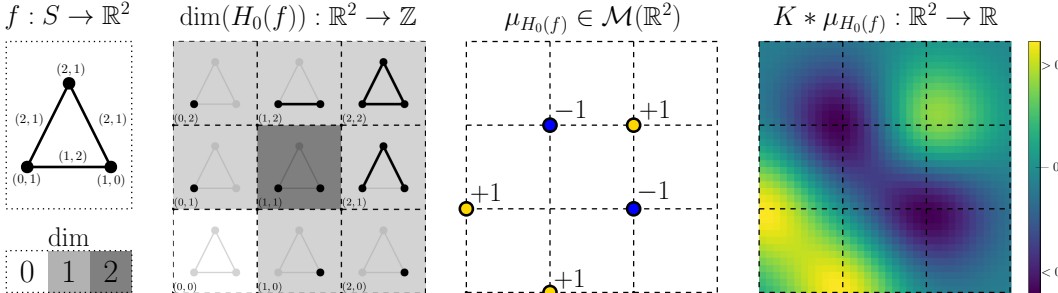

Figure 1: An instance of the pipeline proposed in this article. *Left to right:* A filtered simplicial complex $(S, f)$ (in this case a bi-filtered graph); the Hilbert function of its $0th$ dimensional homology persistence module $H_0(f) : \mathbb{R}^2 \longrightarrow$ vec (which in this case simply counts the number of connected components); the Hilbert decomposition signed measure $\mu_{H_0(f)}$ of the persistence module; and the convolution of the signed measure with a Gaussian kernel.

**Vectorizations.** Viewing signed barcodes as signed point measures enables us to rely on the literature in signed measure theory in order to adapt existing vectorization techniques for usual barcodes to the signed barcodes in a natural way. Our first vectorization (Definition 6) uses convolution with a kernel function and is an adaptation of the *persistence images* of [1]; see Fig. 1. Our second vectorization (Definition 7) uses the notion of slicing of measures of [59] and is an adaptation of the *sliced Wasserstein kernel* of [22]. Both vectorizations are easy to implement and run fast in practice; we assess the runtime of our pipeline in Appendix D. We find that our pipeline is faster than the other topological baselines by one or more orders of magnitude.

**Theoretical stability guarantees.** We prove that our two vectorizations are Lipschitz-continuous with respect to the Kantorovich–Rubinstein norm on signed measures and the norm of the corresponding Hilbert space (Theorems 2 and 3). Combining these results with the Lipschitz-stability of the signed barcodes themselves (Theorem 1) ensures the robustness of our entire feature generation pipeline with respect to perturbations of the filtrations.

**Experimental validation.** Let us emphasize that our choices of signed barcode descriptors are strictly weaker than the usual barcode in the one-parameter case. In spite of this, our experiments show that the performance of the one-parameter version of our descriptors is comparable to that of the usual barcode. We then demonstrate that this good behavior generalizes to the multiparameter case, where it is in fact amplified since our pipeline can outperform its multiparameter competitors (which rely on theoretically stronger descriptors and have been shown to perform already better than the usual one-parameter TDA pipeline) and is competitive with other, non-topological baselines, on a variety of data types (including graphs and time series). For a proof that our descriptors are indeed weaker than other previously considered descriptors, we refer the reader to Proposition 2 in Appendix A.

## 1.3 Related work

We give a brief overview of related vectorization methods for multiparameter persistence; we refer the reader to Appendix B for more details.

The *multiparameter persistence kernel* [32] and *multiparameter persistence landscape* [76] restrict the multiparameter persistence module to certain families of one-parameter lines and leverage some of the available vectorizations for one-parameter barcodes. The *generalized rank invariant landscape* [79] computes the generalized rank invariant over a prescribed collection of intervals (called *worms*) then, instead of decomposing the invariant as we do, it stabilizes it using ideas comings from persistence landscapes [13]. The *multiparameter persistence image* [20] decomposes the multiparameter persistence module into interval summands and vectorizes these summands individually. Since this process is known to be unstable, there is no guarantee on the stability of the corresponding vectorization. The *Euler characteristic surfaces* [6] and the methods of [42] do not work at the level of persistence modules but rather at the level of filtered simplicial complexes, and are based on the computation of the Euler characteristic of the filtration at each index in the multifiltration. These methods are very efficient when the filtered simplicial complexes are small, but

can be prohibitively computationally expensive for high-dimensional simplicial complexes, such as Vietoris–Rips complexes (Example 2).

## 2 Background

In this section, we recall the basics on multiparameter persistent homology and signed measures. We let vec denote the collection of finite dimensional vector spaces over a fixed field $\Bbbk$. Given $n \geq 1 \in \mathbb{N}$, we consider $\mathbb{R}^n$ as a poset, with $x \leq y \in \mathbb{R}^n$ if $x_i \leq y_i$ for all $1 \leq i \leq n$.

**Simplicial complexes.** A finite *simplicial complex* consists of a finite set $S_0$ together with a set $S$ of non-empty subsets of $S_0$ such that, if $s \in S_0$ then $\{s\} \in S$, and if $\tau_2 \in S$ and $\emptyset \neq \tau_1 \subseteq \tau_2$, then $\tau_1 \in S$. We denote such a simplicial complex by $S$, refer to the elements of $S$ as the *simplices* of $S$, and refer to $S_0$ as the *underlying set* of $S$. The *dimension* of a simplex $\tau \in S$ is $\dim(\tau) = |\tau| - 1 \in \mathbb{N}$. In particular, the simplices of dimension 0 correspond precisely to the elements of the underlying set.

**Example 1.** Let $G$ be a simple, undirected graph. Then $G$ can be encoded as a simplicial complex $S$ with simplices only of dimensions 0 and 1, by letting the underlying set of $S$ be the vertex set of $G$, and by having a simplex of dimension 0 (resp. 1) for each vertex (resp. edge) of $G$.

**Definition 1.** A *(multi)filtered simplicial complex* is a pair $(S, f)$ with $S$ a simplicial complex and $f : S \longrightarrow \mathbb{R}^n$ a monotonic map, i.e., a function such that $f(\tau_1) \leq f(\tau_2)$ whenever $\tau_1 \subseteq \tau_2 \in S$. Given a filtered simplicial complex $(S, f : S \longrightarrow \mathbb{R}^n)$ and $x \in \mathbb{R}^n$, we get a subcomplex $S_x^f := \{\tau \in S : f(\tau) \leq x\} \subseteq S$, which we refer to as the *$x$-sublevel set* of $(S, f)$.

**Example 2.** Let $(P, d_P)$ be a finite metric space. The *(Vietoris–)Rips complex* of $P$ is the filtered simplicial complex $(S, f : S \longrightarrow \mathbb{R})$, where $S$ is the simplicial complex with underlying set $P$ and all non-empty subsets of $P$ as simplices, and $f(\{p_0, \ldots, p_n\}) = \max_{i,j} d_P(p_i, p_j)$. Then, for example, given any $x \in \mathbb{R}$, the connected components of the $x$-sublevel set $S_x^f$ coincide with the single-linkage clustering of $P$ at distance scale $x$ (see, e.g., [18]). In many applications, it is useful to also filter the Rips complex by an additional function $d : P \longrightarrow \mathbb{R}$ (such as a density estimator). The *function-Rips complex* of $(P, d)$ is the filtered simplicial complex $(S, g : S \longrightarrow \mathbb{R}^2)$ with $g(\{p_0, \ldots, p_n\}) = (f(\{p_0, \ldots, p_n\}), -\min_i d(p_i))$. Thus, for example, the $(t, -u)$-sublevel set is the Rips complex at distance scale $t$ of the subset of $P$ of points with $d$-value at least $u$. See [17, 19, 25] for examples.

**Homology.** For a precise definition of homology with visual examples, see, e.g., [40]; the following will be enough for our purposes. Given $i \in \mathbb{N}$, the *homology* construction maps any finite simplicial complex $S$ to a $\Bbbk$-vector space $H_i(S)$ and any inclusion of simplicial complexes $S \subseteq R$ to a $\Bbbk$-linear map $H_i(S \subseteq R) : H_i(S) \longrightarrow H_i(R)$. Homology is functorial, meaning that, given inclusions $S \subseteq R \subseteq T$, we have an equality $H_i(R \subseteq T) \circ H_i(S \subseteq R) = H_i(S \subseteq T)$.

**Multiparameter persistence modules.** An *$n$-parameter persistence module* consists of an assignment $M : \mathbb{R}^n \longrightarrow$ vec together with, for every pair of comparable elements $x \leq y \in \mathbb{R}^n$, a linear map $M(x \leq y) : M(x) \longrightarrow M(y)$, with the property that $M(x \leq x)$ is the identity map for all $x \in \mathbb{R}^n$, and that $M(y \leq z) \circ M(x \leq y) = M(x \leq z)$ for all $x \leq y \leq z \in \mathbb{R}^n$.

**Example 3.** Let $(S, f : S \longrightarrow \mathbb{R}^n)$ be a filtered simplicial complex. Then, we have an inclusion $S_x^f \subseteq S_y^f$ whenever $x \leq y$. By functoriality, given any $i \in \mathbb{N}$, we get an $n$-parameter persistence module $H_i(f) : \mathbb{R}^n \longrightarrow$ vec by letting $H_i(f)(x) := H_i(S_x^f)$.

**Definition 2.** Let $M : \mathbb{R}^n \longrightarrow$ vec. The *Hilbert function* of $M$, also known as the *dimension vector* of $M$, is the function $\dim(M) : \mathbb{R}^n \longrightarrow \mathbb{Z}$ given by $\dim(M)(x) = \dim(M(x))$.

In practice, one often deals with persistence modules defined on a finite grid of $\mathbb{R}^n$. These persistence modules are *finitely presentable* (fp), which, informally, means that they can be encoded using finitely many matrices. See Definition 8 in the appendix for a formal definition, but note that this article can be read without a precise understanding of this notion.

**Signed measures.** The Kantorovich–Rubinstein norm was originally defined by Kantorovich and subsequently studied in [46] in the context of measures on a metric space; see [43]. We denote by $\mathcal{M}(\mathbb{R}^n)$ the space of finite, signed, Borel measures on $\mathbb{R}^n$, and by $\mathcal{M}_0(\mathbb{R}^n) \subseteq \mathcal{M}(\mathbb{R}^n)$ the subspace of measures of total mass zero. For $\mu \in \mathcal{M}(\mathbb{R}^n)$, we let $\mu = \mu^+ - \mu^-$ denote its Jordan decomposition [8, p. 421], so that $\mu^+$ and $\mu^-$ are finite, positive measures on $\mathbb{R}^n$.

**Definition 3.** For $p \in [1, \infty]$, the *Kantorovich–Rubinstein norm* of $\mu \in \mathcal{M}_0(\mathbb{R}^n)$ is defined as

$$\|\mu\|_p^{\mathsf{KR}} = \inf \left\{ \int_{\mathbb{R}^n \times \mathbb{R}^n} \|x - y\|_p \, d\psi(x, y) \ : \ \psi \text{ is a measure on } \mathbb{R}^n \times \mathbb{R}^n \text{ with marginals } \mu^+, \mu^- \right\}.$$

We can then compare any two measures $\mu, \nu \in \mathcal{M}(\mathbb{R}^n)$ of the same total mass using $\|\mu - \nu\|_p^{\mathsf{KR}}$.

**Remark 1.** Note that, if $p \geq q$, then $\| - \|_p^{\mathsf{KR}} \leq \| - \|_q^{\mathsf{KR}}$. For $n = 1$, the definition of $\| - \|_p^{\mathsf{KR}}$ is independent of $p$, and in that case we just denote it by $\| - \|^{\mathsf{KR}}$.

Given $x \in \mathbb{R}^n$, let $\delta_x$ denote the Dirac measure at $x$. A *finite signed point measure* is any measure in $\mathcal{M}(\mathbb{R}^n)$ that can be written as a finite sum of signed Dirac measures.

The following result says that, for point measures, the computation of the Kantorovich–Rubinstein norm reduces to an assignment problem, and that, in the case of point measures on the real line, the optimal assignment has a particularly simple form.

**Proposition 1.** *Let $\mu \in \mathcal{M}_0(\mathbb{R}^n)$ be a finite signed point measure with $\mu^+ = \sum_i \delta_{x_i}$ and $\mu^- = \sum_i \delta_{y_i}$, where $X = \{x_1, \ldots, x_k\}$ and $Y = \{y_1, \ldots, y_k\}$ are lists of points of $\mathbb{R}^n$. Then,*

$$\|\mu\|_p^{\mathsf{KR}} = \min \left\{ \sum_{1 \leq i \leq k} \|x_i - y_{\gamma(i)}\|_p \ : \ \gamma \text{ is a permutation of } \{1, \ldots, k\} \right\}.$$

*Moreover, if $n = 1$ and $X$ and $Y$ are such that $x_i \leq x_{i+1}$ and $y_i \leq y_{i+1}$ for all $1 \leq i \leq k - 1$, then the above minimum is attained at the identity permutation $\gamma(i) = i$.*

## 3 Signed barcodes as measures and stable vectorizations

In this section, we introduce the signed barcodes and measures associated to multifiltered simplicial complexes, as well as our proposed vectorizations, and we prove their stability properties.

The barcode [41] of a one-parameter persistence module consists of a multiset of intervals of $\mathbb{R}$, often thought of as a positive integer linear combination of intervals, referred to as bars. This positive integer linear combination can, in turn, be represented as a point measure in the extended plane by recording the endpoints of the intervals [36]. Recent adaptations of the notion of one-parameter barcode to multiparameter persistence [4, 11, 12, 49] assign, to each multiparameter persistence module $M : \mathbb{R}^n \longrightarrow \text{vec}$, *two* multisets $(\mathcal{B}^+, \mathcal{B}^-)$ of bars; each bar typically being some connected subset of $\mathbb{R}^n$. In [11, 55], these pairs of multisets of bars are referred to as *signed barcodes*. In several cases, the multisets $\mathcal{B}^+$ and $\mathcal{B}^-$ are disjoint, and can thus be represented without loss of information as integer linear combinations of bars. This is the case for the minimal Hilbert decomposition signed barcode of [55], where each bar is of the form $\{y \in \mathbb{R}^n : y \geq x\} \subseteq \mathbb{R}^n$ for some $x \in \mathbb{R}^n$. The minimal Hilbert decomposition signed barcode of a multiparameter persistence module $M$ can thus be encoded as a signed point measure $\mu_M$, by identifying the bar $\{y \in \mathbb{R}^n : y \geq x\}$ with the Dirac measure $\delta_x$ (see Fig. 1).

In Section 3.1, we give a self-contained description of the signed measure associated to the minimal Hilbert decomposition signed barcode, including optimal transport stability results for it. Then, in Section 3.2, we describe our proposed vectorizations of signed measures and their stability. The proofs of all of the results in this section can be found in Appendix A.

### 3.1 Hilbert and Euler decomposition signed measures

We start by interpreting the minimal Hilbert decomposition signed barcode of [55] as a signed measure. We prove that this notion is well-defined in Appendix A.4, and give further motivation in Appendix A.5.

**Definition 4.** The *Hilbert decomposition signed measure* of a fp multiparameter persistence module $M : \mathbb{R}^n \longrightarrow \text{vec}$ is the unique signed point measure $\mu_M \in \mathcal{M}(\mathbb{R}^n)$ with the property that

$$\dim(M(x)) = \mu_M\big(\{y \in \mathbb{R}^n : y \leq x\}\big), \quad \text{for all } x \in \mathbb{R}^n.$$

Given a filtered simplicial complex, one can combine the Hilbert decomposition signed measure of all of its homology modules as follows.

**Definition 5.** The *Euler decomposition signed measure* of a filtered simplicial complex $(S, f)$ is

$$\mu_{\chi(f)} := \sum_{i \in \mathbb{N}} (-1)^i \, \mu_{H_i(f)}.$$

The next result follows from [9, 55], and ensures the stability of the signed measures introduced above. In the result, we denote $\|h\|_1 = \sum_{\tau \in S} \|h(\tau)\|_1$ when $(S, h)$ is a filtered simplicial complex.

**Theorem 1.** *Let $n \in \mathbb{N}$, let $S$ be a finite simplicial complex, and let $f, g : S \longrightarrow \mathbb{R}^n$ be monotonic.*

 1. *For $n \in \{1, 2\}$ and $i \in \mathbb{N}$, $\|\mu_{H_i(f)} - \mu_{H_i(g)}\|_1^{\mathsf{KR}} \leq n \cdot \|f - g\|_1$.*

 2. *For all $n \in \mathbb{N}$, $\|\mu_{\chi(f)} - \mu_{\chi(g)}\|_1^{\mathsf{KR}} \leq \|f - g\|_1$.*

Extending Theorem 1 (1.) to a number of parameters $n > 2$ is an open problem.

**Computation.** The design of efficient algorithms for multiparameter persistence is an active area of research [53, 48]. The worst case complexity for the computation of the Hilbert function of the $i$th persistent homology module is typically in $O\left((|S_{i-1}| + |S_i| + |S_{i+1}|)^3\right)$, where $|S_k|$ is the number of $k$-dimensional simplices of the filtered simplicial complex [53, Rmk. 4.3]. However, in one-parameter persistence, the computation is known to be almost linear in practice [5]. For this reason, our implementation reduces the computation of Hilbert functions of homology multiparameter persistence modules to one-parameter persistence; we show in Appendix D.1 that this scalable in practice. We consider persistence modules and their corresponding Hilbert functions restricted to a grid $\{0, \ldots, m - 1\}^n$. By [55, Rmk. 7.4], given the Hilbert function $\{0, \ldots, m - 1\}^n \longrightarrow \mathbb{Z}$ of a module $M$ on a grid $\{0, \ldots, m - 1\}^n$, one can compute $\mu_M$ in time $O(n \cdot m^n)$. Using the next result, the Euler decomposition signed measure is computed in linear time in the size of the complex.

**Lemma 1.** *If $(S, f)$ is a filtered simplicial complex, then $\mu_{\chi(f)} = \sum_{\tau \in S} (-1)^{\dim(\tau)} \, \delta_{f(\tau)}$.*

## 3.2 Vectorizations of signed measures

Now that we have established the stability properties of signed barcodes and measures, we turn the focus on finding vectorizations of these representations. We generalize two well-known vectorizations of single-parameter persistence barcodes, namely the *persistence image* [1] and the *sliced Wasserstein kernel* [22]. The former is defined by centering functions around barcode points in the Euclidean plane, while the latter is based on computing and sorting the point projections onto a fixed set of lines. Both admits natural extensions to points in $\mathbb{R}^n$, which we now define. We also state robustness properties in both cases.

### 3.2.1 Convolution-based vectorization

A *kernel function* is a map $K : \mathbb{R}^n \longrightarrow \mathbb{R}_{\geq 0}$ with $\int_{x \in \mathbb{R}^n} K(x)^2 \, dx < \infty$. Given such a kernel function and $y \in \mathbb{R}^n$, let $K_y : \mathbb{R}^n \longrightarrow \mathbb{R}_{\geq 0}$ denote the function $K_y(x) = K(x - y)$.

**Definition 6.** Given $\mu \in \mathcal{M}(\mathbb{R}^n)$, define the *convolution* of the measure $\mu$ with the kernel function $K$ as $K * \mu \in L^2(\mathbb{R}^n)$ as $(K * \mu)(x) := \int_{z \in \mathbb{R}^n} K(x - z) d\mu(z)$.

**Theorem 2.** *Let $K : \mathbb{R}^n \longrightarrow \mathbb{R}$ be a kernel function for which there exists $c > 0$ such that $\|K_y - K_z\|_2 \leq c \cdot \|y - z\|_2$ for all $y, z \in \mathbb{R}^n$. Then, if $\mu, \nu \in \mathcal{M}(\mathbb{R}^n)$ have the same total mass,*

$$\|K * \mu - K * \nu\|_2 \leq c \cdot \|\mu - \nu\|_2^{\mathsf{KR}}.$$

In Propositions 4 and 5 of the appendix, we show that Gaussian kernels and kernels that are Lipschitz with compact support satisfy the assumptions of Theorem 2.

**Computation.** For the experiments, given a signed measure $\mu$ on $\mathbb{R}^n$ associated to a multiparameter persistence module defined over a finite grid in $\mathbb{R}^n$, we evaluate $K * \mu$ on the same finite grid. As kernel $K$ we use a Gaussian kernel.

### 3.2.2 Sliced Wasserstein kernel

Given $\theta \in S^{n-1} = \{x \in \mathbb{R}^n : \|x\|_2 = 1\} \subseteq \mathbb{R}^n$, let $L(\theta)$ denote the line $\{\lambda \theta : \lambda \in \mathbb{R}\}$, and let $\pi^\theta : \mathbb{R}^n \longrightarrow \mathbb{R}$ denote the composite of the orthogonal projection $\mathbb{R}^n \longrightarrow L(\theta)$ with the map $L(\theta) \longrightarrow \mathbb{R}$ sending $\lambda \theta$ to $\lambda$.

**Definition 7.** Let $\alpha$ be a measure on $S^{n-1}$. Let $\mu, \nu \in \mathcal{M}(\mathbb{R}^n)$ have the same total mass. Their *sliced Wasserstein distance* and *sliced Wasserstein kernel* with respect to $\alpha$ are defined by

$$SW^\alpha(\mu, \nu) := \int_{\theta \in S^{n-1}} \left\| \pi_*^\theta \mu - \pi_*^\theta \nu \right\|^{\mathsf{KR}} d\alpha(\theta) \quad \text{and} \quad k_{SW}^\alpha(\mu, \nu) := \exp\left( - SW^\alpha(\mu, \nu) \right),$$

respectively, where $\pi_*^\theta$ denotes the pushforward of measures along $\pi^\theta : \mathbb{R}^n \longrightarrow \mathbb{R}$.

**Theorem 3.** *Let $\alpha$ be a measure on $S^{n-1}$. If $\mu, \nu \in \mathcal{M}(\mathbb{R}^n)$ have the same total mass, then $SW^\alpha(\mu, \nu) \leq \alpha(S^{n-1}) \cdot \|\mu - \nu\|_2^{\mathsf{KR}}$. Moreover, there exists a Hilbert space $\mathcal{H}$ and a map $\Phi_{SW}^\alpha : \mathcal{M}_0(\mathbb{R}^n) \longrightarrow \mathcal{H}$ such that, for all $\mu, \nu \in \mathcal{M}_0(\mathbb{R}^n)$, we have $k_{SW}^\alpha(\mu, \nu) = \langle \Phi_{SW}^\alpha(\mu), \Phi_{SW}^\alpha(\nu) \rangle_{\mathcal{H}}$ and $\|\Phi_{SW}^\alpha(\mu) - \Phi_{SW}^\alpha(\nu)\|_{\mathcal{H}} \leq 2 \cdot SW^\alpha(\mu, \nu)$.*

**Computation.** We discretize the computation of the sliced Wasserstein kernel by choosing $d$ directions $\{\theta_1, \ldots, \theta_d\} \subseteq S^{n-1}$ uniformly at random and using as measure $\alpha$ the uniform probability measure with support that sample, scaled by a parameter $1/\sigma$, as is common in kernel methods. The Kantorovich–Rubinstein norm in $\mathbb{R}$ is then computed by sorting the point masses, using Proposition 1.

## 4 Numerical experiments

In this section, we compare the performance of our method against several topological and standard baselines from the literature. We start by describing our methodology (see Appendix C.1 for details about hyperparameter choices). An implementation of our vectorization methods is publicly available at https://github.com/DavidLapous/multipers, and will be eventually merged as a module of the Gudhi library [70].

**Notations for descriptors and vectorizations.** In the following, we use different acronyms for the different versions of our pipeline. H is a shorthand for the Hilbert function, while E stands for the Euler characteristic function. Their corresponding decomposition signed measures are written HSM and ESM, respectively. The sliced Wasserstein kernel and the convolution-based vectorization are denoted by SW and C, respectively. 1P refers to one-parameter, and MP to multiparameter. Thus, for instance, MP-HSM-SW stands for the multi-parameter version of our pipeline based on the Hilbert signed measure vectorized using the sliced Wasserstein kernel. The notations for the methods we compare against are detailed in each experiment, and we refer the reader to Section 1.3 for their description.

**Discretization of persistence modules.** In all datasets, samples consist of filtered simplicial complexes. Given such a filtered simplicial complex $f : S \longrightarrow \mathbb{R}^n$, we consider its homology persistence modules $H_i(f)$ with coefficients in a finite field, for $i \in \{0, 1\}$. We fix a grid size $k$ and a $\beta \in (0, 1)$. For each $1 \leq j \leq n$, we take $r_0^j \leq \cdots \leq r_{k-1}^j \subseteq \mathbb{R}$ uniformly spaced, with $r_0^j$ and $r_{k-1}^j$ the $\beta$ and $1 - \beta$ percentiles of $f_j(S_0) \subseteq \mathbb{R}$, respectively. We then restrict each module $M$ to the grid $\{r_0^1, \ldots, r_{k-1}^1, r_k^1\} \times \cdots \times \{r_0^n, \ldots, r_{k-1}^n, r_k^n\}$ where $r_k^j = 1.1 \cdot (r_{k-1}^j - r_0^j) + r_0^j$, setting $M(x_1, \ldots, x_n) = 0$ whenever $x_j = r_j^k$ for some $j$ in order to ensure that $\mu_M$ has total mass zero.

**Classifiers.** We use an XGBoost classifier [27] with default parameters, except for the kernel methods, for which we use a kernel SVM with regularization parameter in $\{0.001, 0.01, 1, 10, 100, 1000\}$.

### 4.1 Hilbert decomposition signed measure vs barcode in one-parameter persistence

As proven in Proposition 2, in Appendix A, the Hilbert decomposition signed measure is a theoretically weaker descriptor than the barcode. Nevertheless, we show in this experiment that, in the one-parameter case, our pipeline performs as well as the following well known vectorization methods based on the barcode: the persistence image (PI) [1], the persistence landscape (PL) [13], and the sliced Wasserstein kernel (SWK) [22]. We also compare against the method pervec (PV) of [15], which consists of a histogram constructed with all the endpoints of the barcode. It is pointed out in [15] that methods like pervec, which do not use the full information of the barcode, can perform very well in practice. This observation was one of the initial motivations for our work. The results of running these pipelines on some of the point cloud and graph data detailed below are in Table 1. See Appendix C.2 for the details about this experiment. As one can see from the results, signed barcode scores are always located between PV and (standard) barcode scores, which makes sense given that they encode more information than PV, but less than barcodes. The most interesting part, however, is

that the Hilbert decomposition signed barcode scores are always of the same order of magnitude (and sometimes even better) than barcode scores.

## 4.2 Classification of point clouds from time series

We perform time series classification on datasets from the UCR archive [39] of moderate sizes; we use the train/test splits which are given in the archive. These datasets have been used to assess the performance of various topology-based methods in [20]; in particular, they show that multiparameter persistence descriptors outperform one-parameter descriptors in almost all cases. For this reason, we compare only against other multiparameter persistence descriptors: multiparameter persistence landscapes (MP-L), multiparameter persistence images (MP-I), multiparameter persistence kernel (MP-K), and the Hilbert function directly used as a vectorization (MP-H). Note that, in this example, we are not interested in performing point cloud classification up to isometry due to the presence of outliers. We use the numbers reported in [20, Table 1]. We also compare against non-topological, state-of-the-art baselines: Euclidean nearest neighbor (B1), dynamic time warping with optimized warping window width (B2), and constant window width (B3), reporting their accuracies from [39]. Following [20], we use a delay embedding with target dimension 3, so that each time series results in a point cloud in $\mathbb{R}^3$. Also following [20], as filtered complex we use an alpha complex filtered by a distance-to-measure (DTM) [24] with bandwidth $0.1$. As one can see from the results in Table 2, MP-HSM-C is quite effective, as it is almost always better than the other topological baselines, and quite competitive with standard baselines.

Interestingly, MP-HSM-SW does not perform too well in this application. We believe that this is due to the fact that the sliced Wasserstein kernel can give too much importance to point masses that are very far away from other point masses; indeed, the cost of transporting a point mass is proportional to the distance it is transported, which can be very large. This seems to be particularly problematic for alpha complexes filtered by density estimates (such as DTM), since some of the simplices of the alpha complex can be adjacent to vertices with very small density, making this simplices appear very late in the filtration, creating point masses in the Hilbert signed measure that are very far away from the bulk of the point masses. This should not be a problem for Rips complexes, due to the redundancy of simplices in Rips complexes: for any set of points in the point cloud, there will eventually be a simplex between them in the Rips complex. In order to test this hypothesis, we run the same experiment but using density to filter a Rips complex instead of an alpha complex (Table 9 in Appendix C.4). We see that, in this case, the sliced Wasserstein kernel does much better, being very competitive with the non-topological baselines.

## 4.3 Classification of graphs

We evaluate our methods on standard graph classification datasets (see [54] and references therein) containing social graphs as well as graphs coming from medical and biological contexts.

Since we want to compare against topological baselines, here we use the accuracies reported in [20] and [79], which use 5 train/test splits to compute accuracy. We compare against multiparameter persistence landscapes (MP-L), multiparameter persistence images (MP-I), multiparameter persistence kernel (MP-K), the generalized rank invariant landscape (GRIL), and the Hilbert and Euler characteristic functions used directly as vectorizations (MP-H and MP-E). We use the same filtrations as reported in [20], so the simplicial complex is the graph (Example 1), which we filter with two parameters: the heat kernel signature [69] with time parameter 10, and the Ricci curvature [64]. As one can see from the results in Table 3, our pipeline compares favorably to topological baselines. Further experiments on the same data but using 10 train/test splits are given in Appendix C.3: they show that we are also competitive with the topological methods of [42] and the state-of-the-art graph classification methods of [75, 80, 81].

## 4.4 Unsupervised virtual screening

In this experiment we show that the distance between our feature vectors in Hilbert space can be used effectively to identify similar compounds (which are in essence multifiltered graphs) in an unsupervised virtual screening task. Virtual screening (VS) is a computational approach to drug discovery in which a library of molecules is searched for structures that are most likely to bind to a given drug target [61]. VS methods typically take as input a query ligand $q$ and a set $L$ of test ligands,

and they return a linear ordering $O(L)$ of $L$, ranking the elements from more to less similar to $q$. VS methods can be supervised or unsupervised; unsupervised methods [67] order the elements of $L$ according to the output of a fixed dissimilarity function between $q$ and each element of $L$, while supervised methods learn a dissimilarity function and require training data [2, 78]. We remark that, in this experiment, we are only interested in the direct metric comparison of multifiltered graphs using our feature vectors and thus only in unsupervised methods.

We interpret molecules as graphs and filter them using the given bond lengths, atomic masses, and bond types. In order to have the methods be unsupervised, we normalize each filtering function using its standard deviation, instead of cross validating different rescalings as in supervised tasks. We use a grid size of $k = 1000$ for all methods, and fix $\sigma$ for the sliced Wasserstein kernel and the bandwidth of the Gaussian kernel for convolution to 1. We assess the performance of virtual screening methods on the Cleves–Jain dataset [28] using the *enrichment factor* ($EF$); details are in Appendix C.5. We report the best results of [67] (in their Table 4), which are used as baseline in the state-of-the-art methods of [34]. We also report the results of [34], but we point out that those are supervised methods. As one can see from the results in Table 4, MP-ESM-C clearly outperforms unsupervised baselines and approaches the performances of supervised ones (despite being unsupervised itself).

# 5   Conclusions

We introduced a provably robust pipeline for processing geometric datasets based on the vectorization of signed barcode descriptors of multiparameter persistent homology modules, with an arbitrary number of parameters. We demonstrated that signed barcodes and their vectorizations are efficient representations of the multiscale topology of data, which often perform better than other featurizations based on multiparameter persistence, despite here only focusing on strictly weaker descriptors. We believe that this is due to the fact that using the Hilbert and Euler signed measures allows us to leverage well-developed vectorization techniques shown to have good performance in one-parameter persistence. We conclude from this that the way topological descriptors are encoded is as important as the discriminative power of the descriptors themselves.

**Limitations.** (1) Our pipelines, including the choice of hyperparameters for our vectorizations, rely on the cross validation of several parameters, which limits the number of possible choices to consider. (2) The convolution-based vectorization method works well when the signed measure is defined over a fine grid, but the performance degrades with coarser grids. This is a limitation of the current version of the method, since convolution in very fine grids does not scale well in the number of dimensions (i.e., in the number of parameters of the persistence module). (3) The sliced Wasserstein vectorization method for signed measures is a kernel method, and thus does not scale well to very large datasets.

**Future work.** There is recent interest in TDA in the *differentiation* of topological descriptors and their vectorizations. Understanding the differentiability of signed barcodes and of our vectorizations could be used to address limitation (1) by optimizing various hyperparameters using a gradient instead of cross validation. Relatedly, (2) could be addressed by developing a neural network layer taking as input signed point measures, which is able to learn a suitable, relatively small data-dependent grid on which to perform the convolutions. For clarity, our choices of signed barcode descriptor and vectorization of signed measures are among the simplest available options, and, although with these basic choices we saw notable improvements when comparing our methodology to state-of-the-art topology-based methods, it will be interesting to see how our proposed pipeline performs when applied with stronger signed barcode descriptors, such as the minimal rank decomposition of [11], or to the decomposition of invariants such as the one of [79]. Relatedly, our work opens up the way for the generalization of other vectorizations from one-parameter persistence to signed barcodes, and for the study of their performance and statistical properties.

**Acknowledgements.** The authors thank the area chair and anonymous reviewers for their insightful comments and constructive suggestions. They also thank Hannah Schreiber for her great help in the implementation of our method. They are grateful to the OPAL infrastructure from Université Côte d'Azur for providing resources and support. DL was supported by ANR grant 3IA Côte d'Azur (ANR-19-P3IA-0002). LS was partially supported by the National Science Foundation through grants CCF-2006661 and CAREER award DMS-1943758. MC was supported by ANR grant TopModel (ANR-23-CE23-0014). SO was partially supported by Inria Action Exploratoire PREMEDIT. This work was carried out in part when MBB and SO were at the Centre for Advanced Study (CAS), Oslo.

| Dataset | SWK | PI | PL | PV | 1P-HSM-SW | 1P-HSM-C |
|---|---|---|---|---|---|---|
| DistalPhalanxOutlineAgeGroup | 73.4 | 66.9 | 68.3 | 66.9 | 70.5 | 70.5 |
| DistalPhalanxOutlineCorrect | 73.6 | 62.3 | 68.5 | 64.5 | 74.6 | 75.4 |
| DistalPhalanxTW | 64.7 | 60.4 | 62.6 | 55.4 | 63.3 | 62.6 |
| COX2 | 78.8(4.0) | 78.6(1.0) | 79.2(3.7) | 78.2(0.8) | 79.2(2.6) | 80.1(1.3) |
| DHFR | 80.8(6.5) | 76.2(3.5) | 78.6(4.6) | 73.5(4.7) | 76.7(5.7) | 76.7(3.4) |
| IMDB-B | 69.6(4.4) | 66.6(3.3) | 65.4(3.3) | 65.8(4.1) | 64.3(4.6) | 63.7(3.6) |

Table 1: Accuracy scores of one-parameter persistence vectorizations on some of the datasets from Tables 2 and 3. The one-parameter version of our signed barcode vectorizations, on the right, performs as well as other topological methods, despite using less topological information.

| Dataset | B1 | B2 | B3 | MP-K | MP-L | MP-I | MP-H | MP-HSM-SW | MP-HSM-C |
|---|---|---|---|---|---|---|---|---|---|
| DistalPhalanxOutlineAgeGroup | 62.6 | 62.6 | **77.0** | 67.6 | 70.5 | 71.9 | 71.2 | 74.1 | 71.2 |
| DistalPhalanxOutlineCorrect | 71.7 | 72.5 | 71.7 | 74.6 | 69.6 | 71.7 | 73.9 | 71.4 | **75.4** |
| DistalPhalanxTW | 63.3 | 63.3 | 59.0 | 61.2 | 56.1 | 61.9 | 60.4 | 62.6 | **67.6** |
| ProximalPhalanxOutlineAgeGroup | 78.5 | 78.5 | 80.5 | 78.0 | 78.5 | 81.0 | 82.4 | 82.9 | 82.4 |
| ProximalPhalanxOutlineCorrect | 80.8 | 79.0 | 78.4 | 78.7 | 78.7 | 81.8 | 82.1 | 77.7 | 82.5 |
| ProximalPhalanxTW | 70.7 | 75.6 | 75.6 | **79.5** | 73.2 | 76.1 | 77.1 | 77.6 | 77.6 |
| ECG200 | **88.0** | **88.0** | 77.0 | 77.0 | 74.0 | 83.0 | 73 | 71.1 | 84.1 |
| ItalyPowerDemand | **95.5** | **95.5** | 95.0 | 80.7 | 78.6 | 79.8 | 80.5 | 79.3 | 77.8 |
| MedicalImages | 68.4 | **74.7** | 73.7 | 55.4 | 55.7 | 60.0 | 56.6 | 53.3 | 56.2 |
| Plane | 96.2 | **100.0** | **100.0** | 92.4 | 84.8 | 97.1 | 99 | 91.4 | **100** |
| SwedishLeaf | 78.9 | **84.6** | 79.2 | 78.2 | 64.6 | 83.8 | 79 | 66.2 | 79.8 |
| GunPoint | 91.3 | 91.3 | 90.7 | 88.7 | 94.0 | 90.7 | 89.3 | 88.7 | **94.1** |
| GunPointAgeSpan | 89.9 | 96.5 | 91.8 | 93.0 | 85.1 | 90.5 | 91.8 | 87.7 | **96.7** |
| GunPointMaleVersusFemale | 97.5 | 97.5 | **99.7** | 96.8 | 88.3 | 95.9 | 93.7 | 83.5 | 96.8 |
| GunPointOldVersusYoung | 95.2 | 96.5 | 83.8 | 99.0 | 97.1 | **100.0** | 99.7 | 99.4 | 99.4 |
| PowerCons | **93.3** | 92.2 | 87.8 | 85.6 | 84.4 | 86.7 | 88.3 | 83.9 | 88.7 |
| SyntheticControl | 88.0 | 98.3 | **99.3** | 50.7 | 60.3 | 60.0 | 55.3 | 49.7 | 61 |

Table 2: Accuracy scores of baselines and multiparameter persistence methods on time series datasets. Boldface indicates best accuracy and underline indicates best accuracy among topological methods. The convolution-based vectorization with the Hilbert decomposition signed measure (MP-HSM-C) performs very well in comparison to other multiparameter persistence vectorizations, and is competitive with the non-topological baselines.

| Dataset | MP-K | MP-L | MP-I | GRIL | MP-H | MP-E | MP-HSM-SW | MP-ESM-SW | MP-HSM-C | MP-ESM-C |
|---|---|---|---|---|---|---|---|---|---|---|
| COX2 | **79.9(1.8)** | 79.0(3.3) | 77.9(2.7) | 79.8(2.9) | 78.2(2) | 78.4(2.2) | 78.4(0.7) | 78.2(0.4) | 77.1(3) | 78.2(1.5) |
| DHFR | 81.7(1.9) | 79.5(2.3) | 80.2(2.2) | 77.6(2.5) | 81.6(1.6) | 79.6(1.9) | 80(1.1) | 80.8(3) | **81.9(2.5)** | 80.5(3.1) |
| IMDB-B | 68.2(1.2) | 71.2(2.0) | 71.1(2.1) | 65.2(2.6) | 72.3(2.4) | 71.3(1.7) | 72.9(2.1) | 74.7(1.6) | **74.8(2.5)** | 74.4(2.4) |
| IMDB-M | 46.9(2.6) | 46.2(2.3) | 46.7(2.7) | NA | 47(2.7) | 47.3(1.9) | 47.2(2.5) | 47.7(2.4) | **47.9(3.2)** | 47.3(3.2) |
| MUTAG | 86.1(5.2) | 84.0(6.8) | 85.6(7.3) | 87.8(4.2) | 86.7(5.5) | **88.8(4.2)** | 87.3(5) | 87.2(2.6) | 85.6(5.3) | 88.3(5.8) |
| PROTEINS | 67.5(3.1) | 65.8(3.3) | 67.3(3.5) | 70.9(3.1) | 67.4(2.2) | 70.2(5.7) | 72(3.1) | 68.8(2.7) | **74.6(2.1)** | 70.9(0.8) |

Table 3: Accuracy and standard deviation scores of topological methods (averaged over 5-fold train/test splits) on graph datasets. Bold indicates best accuracy. Again, the convolution-based vectorization with the Hilbert decomposition signed measure (MP-HSM-C) performs very well when compared to other multiparameter persistence vectorizations.

| Model | $EF_2$ (max. 50) | $EF_5$ (max. 20) | $EF_{10}$ (max. 10) |
|---|---|---|---|
| USR + GZD | 13.7 | 7.7 | 4.7 |
| USR + PS | 13.1 | 7.9 | 5.0 |
| USR + ROCS | 17.1 | 9.1 | 5.4 |
| GZD + PS | 16.0 | 9.1 | 5.9 |
| GZD + ROCS | 20.3 | 10.8 | 5.3 |
| PSk + ROCS | 20.5 | 10.7 | 6.4 |
| MP-H | 16.0 | 9.0 | 5.8 |
| MP-E | 21.3 | 11.8 | 7.0 |
| MP-ESM-SW | 17.4 | 8.7 | 4.9 |
| MP-HSM-SW | 22.5 | 11.1 | 6.4 |
| MP-HSM-C | 25.4 | 13.3 | 7.6 |
| MP-ESM-C | **28.2** | **13.7** | **8.0** |
| ToDD-RF (supervised) | 35.2(2.3) | 15.6(1.0) | 8.1(0.4) |
| ToDD-ViT (supervised) | 39.6(1.4) | 18.6(0.4) | 9.9(0.1) |

Table 4: Enrichment factor ($EF$) for $\alpha \in \{2, 5, 10\}$ of virtual screening methods on Cleves–Jain data. Bold indicates best $EF$ among unsupervised methods. The convolution-based vectorization of the Euler decomposition signed measure (MP-ESM-C) performs significantly better than unsupervised baselines. Moreover, the pipelines based on the vectorization of signed barcodes perform better than those using the raw Hilbert and Euler functions directly.

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

Appendix to

# Stable Vectorization of Multiparameter Persistent Homology using Signed Barcodes as Measures



**David Loiseaux[1]\*, Luis Scoccola[2]\*, Mathieu Carrière[1], Magnus B. Botnan[3], Steve Oudot[4]**

[1]DataShape, Centre Inria d'Université Côte d'Azur     [2]Mathematics, University of Oxford
[3]Mathematics, Vrije Universiteit Amsterdam     [4]GeomeriX, Inria Saclay and École polytechnique

## Table of Contents

## A   Missing definitions and proofs

For an introduction to multiparameter persistence, see, e.g., [10]. We mostly follow the conventions from [12, 55].

### A.1   Finitely presentable modules

Let $x \in \mathbb{R}^n$. We denote by $P_x : \mathbb{R}^n \longrightarrow \mathrm{vec}$ the persistence module such that $P_x(y) = \Bbbk$ if $y \geq x$ and $P_x(y) = 0$ otherwise, with all structure morphisms that are not forced to be zero being the identity $\Bbbk \longrightarrow \Bbbk$.

---

\* Equal contribution.

**Definition 8.** A persistence module is *finitely generated projective* if it is a finite direct sum of modules of the form $P_x$. A persistence module is *finitely presentable* if it is isomorphic to the cokernel of a morphism between two finitely generated projective modules.

The intuition behind the notion of finitely presentable persistence module is the following. By definition, any such module is isomorphic to the cokernel of a morphism $\bigoplus_{j \in J} P_{y_j} \longrightarrow \bigoplus_{i \in I} P_{x_i}$. The elements $1 \in \Bbbk = P_{x_i}(x_i)$ are often referred to as the generators of the persistence module—and in TDA they correspond to topological features being born—while the elements $1 \in \Bbbk = P_{y_j}(y_j)$ are referred to as the relations—and in TDA they correspond to topological features dying or merging.

It is worth mentioning that most construction related to TDA produce finitely presentable modules:

**Lemma 2.** *If $(S, f : S \longrightarrow \mathbb{R}^n)$ is a finite filtered simplicial complex, then $H_i(f)$ is finitely presentable for all $i \in \mathbb{N}$.*

*Proof.* Let $S_k$ denote the set of $k$-dimensional simplices of $S$. By definition of homology, the persistence module $H_i(f)$ is the cokernel of a morphism $\bigoplus_{\tau \in S_{i+1}} P_{f(\tau)} \longrightarrow K$, where $K$ is the kernel of a morphism $\bigoplus_{\tau \in S_i} P_{f(\tau)} \longrightarrow \bigoplus_{\tau \in S_{i-1}} P_{f(\tau)}$. Since $K$ is the kernel of a morphism between finitely presentable module, it is itself finitely presentable (see, e.g., [9, Lemma 3.14]), it follows that $H_i(f)$ is finitely presentable, since it is the cokernel of a morphism between finitely presentable modules (see, e.g., [51, Exercise 4.8]). $\square$

**Example 4.** Let $a < b \in \mathbb{R}$. Let $\Bbbk_{[a,b)} : \mathbb{R} \longrightarrow \mathrm{vec}$ be the one-parameter persistence module such that $\Bbbk_{[a,b)}(x) = \Bbbk$ if $x \in [a, b)$ and $\Bbbk_{[a,b)}(x) = 0$ if $x \notin [a, b)$; and, if $a \leq x \leq y < b$, then $\Bbbk_{[a,b)}(x \leq y)$ is the identity linear map $\Bbbk \longrightarrow \Bbbk$, and is the zero linear map otherwise. The module $\Bbbk_{[a,b)}$ is often referred to as the *interval module* with support $[a, b)$. This interval module is finitely presentable, since it is isomorphic to the cokernel of any non-zero map $P_b \longrightarrow P_a$.

An example of a persistence module that is not finitely presentable is any interval module supported over an open interval, such as $\Bbbk_{(a,b)}$.

## A.2 Discriminative power of descriptors

It is well-known [19, Theorem 12] that the one-parameter barcode is equivalent to the rank invariant, which is a descriptor that readily generalizes to the multiparameter case, as follows.

**Definition 9** ([19])**.** The *rank invariant* of a persistence module $M : \mathbb{R}^n \longrightarrow \mathrm{vec}$ is the function which maps each pair of comparable elements $x \leq y \in \mathbb{R}^n$ to the rank of the linear map $M(x) \longrightarrow M(y)$.

**Proposition 2.** *The Hilbert function is a strictly weaker descriptor than the rank invariant. In particular, the descriptors of [20, 76, 79] are more discriminative than the Hilbert decomposition signed measure.*

*Proof.* The persistence modules $M = \Bbbk_{[0,\infty)}$ and $N = \Bbbk_{[0,1)} \oplus \Bbbk_{[1,\infty)}$ have the same Hilbert function $\mathbb{R} \longrightarrow \mathbb{Z}$, since in both cases the Hilbert function is just the indicator function of the set $[0, \infty)$. However, the rank of the linear map $M(0 \leq 2) : M(0) \longrightarrow M(2)$ is one, while the rank of the linear map $N(0 \leq 2) : N(0) \longrightarrow N(2)$ is zero, proving the first claim. Since the invariants of [20, 76, 79] determine the rank invariant of (at least some) one-parameter slices, it follows that they are strictly more discriminative than the Hilbert function. $\square$

## A.3 References for Proposition 1 of the main article

Both statements are well known. For the first statement, see, e.g., [57, Proposition 2.1], and, for the second one, see, e.g., [57, Remark 2.30].

## A.4 Definition of the Hilbert decomposition signed measure

For intuition about the concepts in this section, and their connection to [55], see Appendix A.5. Here, to be self-contained, we unfold the definitions of [55].

**Lemma 3** (cf. [55, Proposition 5.2]). *Let $M : \mathbb{R}^n \longrightarrow$ vec be finitely presentable. There exists a pair of finitely generated projective modules $(P, Q)$ such that, as functions $\mathbb{R}^n \longrightarrow \mathbb{Z}$, we have*

$$\dim(M) = \dim(P) - \dim(Q).$$

Note that the above decomposition of the Hilbert function is not unique; in fact, there are infinitely many of these decompositions, given by different pairs $(P, Q)$. Nevertheless, as we will show below, they all yield the same, unique, Hilbert decomposition signed measure. For this we will use the following well known fact from geometric measure theory.

**Lemma 4.** *To prove that $\mu = \nu \in \mathcal{M}(\mathbb{R}^n)$, it is enough to show that, for every $x \in \mathbb{R}^n$, we have*

$$\mu\big(\{y \in \mathbb{R}^n : y \leq x\}\big) = \nu\big(\{y \in \mathbb{R}^n : y \leq x\}\big),$$

*Proof.* Since sets of the form $\{y \in \mathbb{R}^n : y \leq x\}$ generate the Borel sigma-algebra, and the measures in $\mathcal{M}(\mathbb{R}^n)$ are necessarily finite, the result follows from a standard application of the $\pi$-$\lambda$ theorem [37, Theorem A.1.4]. For instance, one can follow the proof of [37, Theorem A.1.5], by noting that positivity of the measures is not required for the proof to work. $\square$

The Hilbert decomposition signed measure is built as a sum of signed Dirac measures, one for each summand of $P$ and $Q$ in the decomposition of Lemma 3. The following lemma justifies this insight.

**Lemma 5.** *Let $x \in \mathbb{R}^n$, let $P_x : \mathbb{R}^n \longrightarrow$ vec be the corresponding finitely generated projective module, and let $\delta_x$ be the corresponding Dirac measure. Then $\dim(P_x)(y) = \delta_x(\{w \in \mathbb{R}^n : w \leq y\})$.*

*Proof.* Note that, by definition of $P_x$, we have $\dim(P_x)(y) = 1$ if $x \leq y$ and $\dim(P_x)(y) = 0$ otherwise. Similarly, $\delta_x(\{w : w \leq y\}) = 1$ if $x \leq y$ and $\delta_x(\{w : w \leq y\}) = 0$ otherwise. The result follows. $\square$

We now have the required ingredients to define the Hilbert decomposition signed measure formally. While the statement itself only claims existence and uniqueness, the proof actually builds the measure explicitly as a finite sum of signed Dirac measures (see Eq. (1)), as explained above.

**Proposition 3.** *If $M : \mathbb{R}^n \longrightarrow$ vec is fp, there exists a unique point measure $\mu_M \in \mathcal{M}(\mathbb{R}^n)$ with*

$$\dim(M(x)) = \mu_M\big(\{y \in \mathbb{R}^n : y \leq x\}\big), \ \text{for all } x \in \mathbb{R}^n.$$

*Proof.* Let $(P, Q)$ be as in Lemma 3. Let $\{x_i\}_{i \in I}$ be such that $P \cong \bigoplus_{i \in I} P_{x_i}$ and let $\{y_j\}_{j \in J}$ be such that $Q \cong \bigoplus_{j \in J} P_{y_j}$. To show existence, define the measure

$$\mu_M = \sum_{i \in I} \delta_{x_i} - \sum_{j \in J} \delta_{y_j}. \tag{1}$$

Then

$$\begin{aligned}
\dim(M(z)) &= \dim(P(z)) - \dim(Q(z)) \\
&= |\{i \in I : x_i \leq z\}| - |\{j \in J : y_j \leq z\}| \\
&= \mu_M(\{w \in \mathbb{R}^n : w \leq z\}),
\end{aligned}$$

where in the second equality we used Lemma 5. Uniqueness follows from Lemma 4. $\square$

### A.5 The Hilbert decomposition signed measure as a signed barcode

We clarify the connection between the Hilbert decomposition signed measure and the concept of signed barcode of [55].

Let $M : \mathbb{R}^n \longrightarrow$ vec be finitely presentable. A pair of finitely generated projective modules $(P, Q)$ as in Lemma 3 is what is called a *Hilbert decomposition* of $\dim(M)$ in [55]. Since the modules $P$ and $Q$ are, by assumption, finitely generated projective, there must exist multisets $\{x_i\}_{i \in I}$ and $\{y_j\}_{j \in J}$ of elements of $\mathbb{R}^n$ such that $P \cong \bigoplus_{i \in I} P_{x_i}$ and $Q \cong \bigoplus_{j \in J} P_{y_j}$. A pair of multisets of elements of $\mathbb{R}^n$ is what is called a *signed barcode* in [55]. The intuition is that each element $x$ of $\mathbb{R}^n$ determines its upset $\{y \in \mathbb{R}^n : x \leq y\}$, which coincides with the support of the module $P_x$ (by Lemma 5). Thus, the supports of the summands of $P$ (resp. $Q$) are interpreted as the positive (resp. negative) bars of the signed barcode $(\{x_i\}_{i \in I}, \{y_j\}_{j \in J})$. See Figs. 2 and 3 for illustrations.

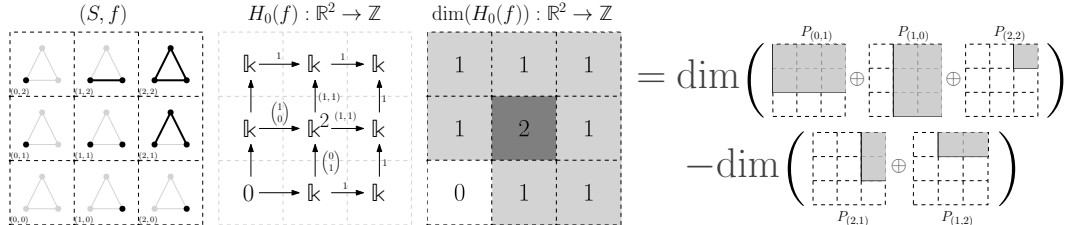

Figure 2: *Left to right:* The filtered simplicial complex of Fig. 1 of the main article; its 0*th* dimensional homology persistence module $H_0(f)$; the Hilbert function of $H_0(f)$; and a decomposition of the Hilbert function of $H_0(f)$ as a linear combination of Hilbert functions of finitely generated projective persistence modules: $\dim(H_0(f)) = \dim(P_{(0,1)} \oplus P_{(1,0)} \oplus P_{(2,2)}) - \dim(P_{(2,1)} \oplus P_{(1,2)})$.

A Hilbert decomposition of $H_0(f)$            Hilbert decomposition signed measure of $H_0(f)$

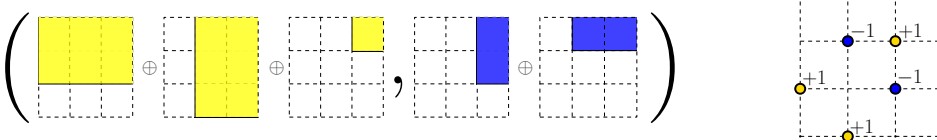

Figure 3: A Hilbert decomposition (in the sense of [55]) of the module of Fig. 2, and the corresponding Hilbert decomposition signed measure. In the Hilbert decomposition, the supports of the persistence modules in yellow are interpreted as positive bars, while the supports of the persistence modules in blue are interpreted as negative bars. Since bars corresponding to finitely generated projective modules are of the form $\{y \in \mathbb{R}^n : y \geq x\}$ for some $x \in \mathbb{R}^n$, they are uniquely characterized by their corresponding $x \in \mathbb{R}^n$; this is why bars in [55] are just taken to be points of $\mathbb{R}^n$.

## A.6 Proof of Theorem 1 of the main article

Claim (1.) follows by combining the stability of sublevel set homology in the presentation distance [9, Theorem 1.9 $(i)$] for $p = 1$ with the stability of bigraded Betti numbers with respect to the presentation distance [55, Theorem 1.5], and using the fact that the signed 1-Wasserstein distance between the signed barcodes of [55] is equal to the distance induced by the Kantorovich–Rubinstein norm between the signed measures associated to the signed barcodes, which is proven in [55, Proposition 6.11].

We now prove claim (2.), using definitions from [55]. Consider the following two signed barcodes (in the sense of [55]):

$$(\mathcal{B}_+, \mathcal{B}_-) := \left( \{f(\tau)\}_{\substack{\tau \in S \text{ s.t.} \\ \dim(\tau) \\ \text{is even}}}, \ \{f(\tau)\}_{\substack{\tau \in S \text{ s.t.} \\ \dim(\tau) \\ \text{is odd}}} \right), \ (\mathcal{C}_+, \mathcal{C}_-) := \left( \{g(\tau)\}_{\substack{\tau \in S \text{ s.t.} \\ \dim(\tau) \\ \text{is even}}}, \ \{g(\tau)\}_{\substack{\tau \in S \text{ s.t.} \\ \dim(\tau) \\ \text{is odd}}} \right)$$

It is clear that the signed measures associated to these signed barcodes, in the sense of [55, Section 6.4], are equal to $\mu_{\chi(f)}$ and $\mu_{\chi(g)}$, respectively. By [55, Proposition 6.11] and [55, Definition 6.1], it is then enough to prove that there exists a bijection $F : \mathcal{B}_+ \cup \mathcal{C}_- \longrightarrow \mathcal{C}_+ \cup \mathcal{B}_-$ such that

$$\sum_{i \in \mathcal{B}_+ \cup \mathcal{C}_-} \|i - F(i)\|_1 \ \leq \ \sum_{\tau \in S} \|f(\tau) - g(\tau)\|_1.$$

To construct such a bijection, we simply map $f(\tau) \in \mathcal{B}_+$ to $g(\tau) \in \mathcal{C}_+$ when $\dim(\tau)$ is even, and $g(\tau) \in \mathcal{C}_-$ to $f(\tau) \in \mathcal{B}_-$ when $\dim(\tau)$ is odd.

We remark that the content of claim (2.) is essentially the same as that of [42, Lemma 12], although they use a slightly different terminology.

## A.7 Proof of Lemma 1 of the main article

It is well-known [72, Theorem 12.4.1] that, if $S$ is a finite simplicial complex, then

$$\sum_{i \in \mathbb{N}} (-1)^i \dim(H_i(S; \mathbb{k})) = \sum_{\tau \in S} (-1)^{\dim(\tau)}.$$

It follows from this and Proposition 3 that, if $(S, f)$ is a filtered simplicial complex, with $f : S \longrightarrow \mathbb{R}^n$, and $x \in \mathbb{R}^n$, then

$$\mu_{\chi(f)}\left(\{y \in \mathbb{R}^n : y \le x\}\right) = \sum_{\substack{\tau \in S \\ f(\tau) \le x}} (-1)^{\dim(\tau)}.$$

Now note that the right-hand side is also equal to the measure of the set $\{y \in \mathbb{R}^n : y \le x\}$ with respect to the signed measure $\sum_{\tau \in S} (-1)^{\dim(\tau)} \delta_{f(\tau)}$. The result then follows from Lemma 4.

### A.8 Lipschitz-stability of vectorizations

*Proof of Theorem 2 of the main article.* Let $\lambda = \mu - \nu$ and let $\psi$ be a coupling between $\lambda^+$ and $\lambda^-$. Then, for every $x \in \mathbb{R}^n$, we have

$$(K * \mu - K * \nu)(x) = (K * \lambda^+ - K * \lambda^-)(x) = \int_{\mathbb{R}^n} K(x - y) \, d\mu(y) - \int_{\mathbb{R}^n} K(x - z) \, d\nu(z),$$

which is equal to $\int_{\mathbb{R}^n \times \mathbb{R}^n} (K(x - y) - K(x - z)) \, d\psi(y, z)$, since $K(x - y)$ does not depend on $z$ and $K(x - z)$ does not depend on $y$. Thus,

$$\begin{aligned}
\|K * \mu - K * \nu\|_2^2 &= \left( \int_{\mathbb{R}^n} \left( \int_{\mathbb{R}^n \times \mathbb{R}^n} (K(x - y) - K(x - z)) \, d\psi(y, z) \right)^2 dx \right)^{1/2} \\
&\le \int_{\mathbb{R}^n \times \mathbb{R}^n} \left( \int_{\mathbb{R}^n} (K(x - y) - K(x - z))^2 \, dx \right)^{1/2} d\psi(y, z) \\
&= \int_{\mathbb{R}^n \times \mathbb{R}^n} \|K_y - K_z\|_2 \, d\psi(y, z) \\
&\le \int_{\mathbb{R}^n \times \mathbb{R}^n} c \, \|y - z\|_2 \, d\psi(y, z),
\end{aligned}$$

where in the first inequality we used Minkowski's integral inequality [44, Theorem 202], and in the second inequality we used the hypothesis about $K$. Since the coupling $\psi$ is arbitrary, we have $\|K * \mu - K * \nu\|_2 \le c \, \|\lambda\|_2^{\mathsf{KR}} = c \, \|\mu - \nu\|_2^{\mathsf{KR}}$, as required. $\qquad\square$

**Proposition 4.** *Let $K : \mathbb{R}^n \longrightarrow \mathbb{R}$ be a Gaussian kernel with zero mean and covariance $\Sigma$. For all $y, z \in \mathbb{R}^n$, we have $\|K_y - K_z\|_2 \le c \, \|y - z\|_2$, with*

$$c = \frac{\|\Sigma^{-1}\|_2^{1/2}}{\sqrt{2} \, \pi^{n/4} \, \det(\Sigma)^{1/4}},$$

*where $\|\Sigma^{-1}\|_2$ denotes the operator norm of $\Sigma^{-1}$ associated to the Euclidean norm.*

*Proof.* By definition,

$$K(x) = \frac{\exp\left(-\frac{1}{2} \|x\|_{\Sigma^{-1}}^2\right)}{\sqrt{(2\pi)^n \det(\Sigma)}},$$

where $\| - \|_{\Sigma^{-1}}$ denotes the norm associated to the quadratic form $\Sigma^{-1}$.

We start by noting that $\|K_y - K_z\|_2^2 = 2\|K\|_2^2 - 2\langle K_y, K_z \rangle$. Now, a standard computation shows that

$$\|K\|_2^2 = \int_{\mathbb{R}^n} \frac{\exp\left(-\|x\|_{\Sigma^{-1}}^2\right)}{(2\pi)^n \det(\Sigma)} \, dx = \frac{1}{\pi^{n/2} \det(\Sigma)^{1/2}}.$$

The term $\langle K_y, K_z \rangle$ can be computed similarly, using the formula for the product of Gaussian densities [56, Section 8.1.8]:

$$
\begin{aligned}
\langle K_y, K_z \rangle &= \int_{\mathbb{R}^n} K_y(x)\, K_z(x)\, dx \\
&= \int_{\mathbb{R}^n} \frac{\exp\left(-\|x - \frac{y+z}{2}\|^2_{\Sigma^{-1}} - \|\frac{y-z}{2}\|^2_{\Sigma^{-1}}\right)}{(2\pi)^n \det(\Sigma)}\, dx \\
&= \exp\left(-\left\|\frac{y-z}{2}\right\|^2_{\Sigma^{-1}}\right) \int_{\mathbb{R}^n} \frac{\exp(-\|x - \frac{y+z}{2}\|^2_{\Sigma^{-1}})}{(2\pi)^n \det(\Sigma)}\, dx \\
&= \frac{\exp\left(-\|\frac{y-z}{2}\|^2_{\Sigma^{-1}}\right)}{\pi^{n/2} \det(\Sigma)^{1/2}}.
\end{aligned}
$$

Thus,

$$
\|K_y - K_z\|^2_2 = 2\|K\|^2_2 - 2\langle K_y, K_z \rangle = 2\frac{1 - \exp\left(-\|\frac{y-z}{2}\|^2_{\Sigma^{-1}}\right)}{\pi^{n/2} \det(\Sigma)^{1/2}} \leq 2\frac{\|\frac{y-z}{2}\|^2_{\Sigma^{-1}}}{\pi^{n/2} \det(\Sigma)^{1/2}},
$$

where, for the inequality, we use the fact that the function $z \mapsto 1 - \exp(-z)$ is 1-Lipschitz when restricted to $\mathbb{R}_{\geq 0}$. The result now follows from the fact that $\|y - z\|^2_{\Sigma^{-1}} \leq \|\Sigma^{-1}\|_2 \|y - z\|^2_2$, by a standard property of the operator norm. $\qquad\square$

**Proposition 5.** *Let $K : \mathbb{R}^n \longrightarrow \mathbb{R}$ be $a$-Lipschitz and of compact support. For all $y, z \in \mathbb{R}^n$, we have $\|K_y - K_z\|_2 \leq c\|y - z\|_2$, with $c = a \cdot (2\,|\mathrm{supp}(K)|)^{1/2}$, where $|\mathrm{supp}(K)|$ denotes the Lebesgue measure of the support of $K$.*

*Proof.* Note that $|K_y(x) - K_z(x)| = |K(x-y) - K(x-z)| \leq a\,\|(x-y) - (x-z)\|_2 = a\,\|y-z\|_2$, by assumption. Now

$$
\begin{aligned}
\|K_y - K_z\|^2_2 &= \int_{\mathbb{R}^n} (K_y(x) - K_z(x))^2\, dx \\
&= \int_{\mathrm{supp}(K_y) \cup \mathrm{supp}(K_z)} (K_y(x) - K_z(x))^2\, dx \\
&\leq \int_{\mathrm{supp}(K_y) \cup \mathrm{supp}(K_z)} a^2\, \|y-z\|^2_2\, dx \\
&= a^2\, \|y-z\|^2_2 \int_{\mathrm{supp}(K_y) \cup \mathrm{supp}(K_z)} dx \\
&\leq a^2 \cdot 2\,|\mathrm{supp}(K)| \cdot \|y-z\|^2_2
\end{aligned}
$$

as required. $\qquad\square$

**Lemma 6.** *Let $\alpha$ be a measure on the $(n-1)$-dimensional sphere $S^{n-1}$. The function $SW^\alpha$ is conditionally negative semi-definite on the set $\mathcal{M}_0(\mathbb{R}^n)$.*

*Proof.* Let $a_1, \ldots, a_k \in \mathbb{R}$ such that $\sum_i a_i = 0$ and let $\mu_1, \ldots, \mu_k \in \mathcal{M}_0(\mathbb{R}^n)$. Let $\theta \in S^{n-1}$, let $\nu_i = \pi^\theta_* \mu_i$, and let $\lambda_i = \nu_i^+ + \sum_{\ell \neq i} \nu_\ell^-$. Note that $\lambda_i$ is a positive measure on $\mathbb{R}$ for all $i$, and that $\lambda_i(\mathbb{R}) = \nu_i^+(\mathbb{R}) + \sum_{\ell \neq i} \nu_\ell^-(\mathbb{R}) = \nu_i^+(\mathbb{R}) + \sum_{\ell \neq i} \nu_\ell^+(\mathbb{R}) = \sum_\ell \nu_\ell^+(\mathbb{R}) = m$ is independent of $i$. Then

$$
\begin{aligned}
\|\nu_i - \nu_j\|^{\mathsf{KR}} &= \left\| \nu_i^+ + \nu_j^- + \sum_{i \neq \ell \neq j} \nu_\ell^- - \left( \nu_j^+ + \nu_i^- + \sum_{i \neq \ell \neq j} \nu_\ell^- \right) \right\|^{\mathsf{KR}} \\
&= \|\lambda_i - \lambda_j\|^{\mathsf{KR}}.
\end{aligned}
$$

It follows that $\sum_{i,j} a_i a_j \|\nu_i - \nu_j\|^{\mathsf{KR}} = \sum_{i,j} a_i a_j \|\lambda_i - \lambda_j\|^{\mathsf{KR}} \leq 0$ since the Wasserstein distance on positive measures of a fixed total mass $m$ on the real line is conditionally negative semi-definite [22, Proposition 2.1 (i)], as it is isometric to an $L^1$-distance [57, Remark 2.30]. The result then follows by integrating over $\theta \in S^{n-1}$. $\qquad\square$

*Proof of Theorem 3.* We start with the stability of the sliced Wasserstein distance. By linearity of integration, it is enough to prove that $\big\| \pi_*^\theta \mu - \pi_*^\theta \nu \big\|^{\mathsf{KR}} \leq \|\mu - \nu\|_2^{\mathsf{KR}}$ for every $\theta \in S^{n-1}$, which follows directly from the fact that orthogonal projection onto $L(\theta)$ is a 1-Lipschitz map when using Euclidean distances.

To prove the existence of a Hilbert space $\mathcal{H}$ and a map $\Phi_{SW}^\alpha : \mathcal{M}_0(\mathbb{R}^n) \longrightarrow \mathcal{H}$ such that, for all $\mu, \nu \in \mathcal{M}_0(\mathbb{R}^n)$, we have $k_{SW}^\alpha(\mu, \nu) = \langle \Phi_{SW}^\alpha(\mu), \Phi_{SW}^\alpha(\nu) \rangle_\mathcal{H}$, we apply [7, Theorem 3.2.2, p.74]; this requires us to prove that $SW^\alpha$ is a conditionally negative semi-definite distance, which we do in Lemma 6. To conclude, we must show that $\|\Phi_{SW}^\alpha(\mu) - \Phi_{SW}^\alpha(\nu)\|_\mathcal{H} \leq 2 \cdot SW^\alpha(\mu, \nu)$. This follows from the fact that $\|\Phi_{SW}^\alpha(\mu) - \Phi_{SW}^\alpha(\nu)\|_\mathcal{H} = 2 - 2 \cdot k_{SW}^\alpha(\mu, \nu)$ and that the function $z \mapsto 1 - \exp(-z)$ is 1-Lipschitz when restricted to $\mathbb{R}_{\geq 0}$. $\square$

# B   Review of numerical descriptors of persistence modules

**Hilbert function.** In the one-parameter case, the Hilbert function is known as the Betti curve; see, e.g., [74], and [73] for an extension beyond the one-parameter case. The Hilbert function is itself a numerical descriptor, so it can be used as a feature to train vector-based machine learning models; it has been used in the multiparameter case in [34], where it performs favorably when compared to application-specific state-of-the-art methods for drug discovery tasks.

**Euler characteristic.** The (pointwise) Euler characteristic of a (multi)filtered simplicial complex is also readily a numerical invariant. It has been used to train machine leaning models in the multiparameter case in [6, 42].

**Barcode-based.** Many numerical invariants of persistence modules based on the one-parameter barcode have been proposed (see, e.g., [3] for a survey). Since these methods rely on the one-parameter barcode, they do not immediately generalize to the multiparameter case.

**Barcode endpoints and filtration values.** It has been shown that the full discriminating power of barcodes is not needed to achieve good performance in topology-based classifications tasks [15, 3]. Indeed, [15] argues that, in many cases, features which only use the endpoints of barcodes, and thus forget the pairing between endpoints, perform as well as features that do use the pairings. The work [3] reaches somewhat similar conclusions, although their descriptors do keep some of the information encoded by the pairing between endpoints given by the barcode (in particular making their descriptors not immediately generalizable to the multiparameter case). The analysis of [15] is particularly relevant to our work: our Hilbert decomposition signed measure can be interpreted as a signed version of their `pervec` descriptor, and our Euler characteristic signed measure can be interpreted as a signed version of their `filvec` descriptor.

**Rank invariant.** The rank invariant (Definition 9) can be encoded as a function $\mathbb{R}^n \times \mathbb{R}^n \longrightarrow \mathbb{Z}$, by declaring the function to be zero on pairs $x \not\leq y$. However, to our knowledge, the rank invariant has not been used directly as a numerical descriptor to train vector-based machine learning models. Vectorizations of the rank invariant, and of some of its generalized versions [49, 11, 4], have been introduced building on the notion of persistence landscape.

**Persistence landscape.** The persistence landscape [13] is a numerical descriptor of one-parameter persistence modules which completely encodes the rank invariant of the module. The persistence landscape was extended to a descriptor of multiparameter persistence modules in [76] by stacking persistence landscapes associated to the restriction of the multiparameter persistence modules to all lines of slope 1. Another extension of the persistence landscape, this time to the 2-parameter case, is the generalized rank invariant landscape [79], which relies on the generalized rank invariant restricted to certain convex shapes in $\mathbb{R}^2$.

**Multiparameter persistence kernel.** Any kernel method for one-parameter persistence modules, such as [60, 50, 22, 52], gives rise to a kernel method for multiparameter persistence modules, using the methodology of [32], which relies on "slicing", that is, on restricting multiparameter persistence modules to monotonic lines in their parameter space.

**Multiparameter persistence image.** Another vectorization method which relies on slicing multiparameter persistence modules is in [20]. Their method uses the notion of vineyard [31] to relate the barcodes of different slices and outputs a descriptor which encodes these relationships.

## C Further details about experiments

### C.1 Hyperparameter choices

We fix $\beta = 0.01$ in all experiments. We also use $d = 50$ slicing lines and a grid size of $k = 1000$ for MP-SW. In supervised learning tasks, all parameters are chosen using 10-fold cross validation (cv). Beside the kernel SVM regularization parameter, these include: the size of the grid $k$ is chosen to be in $\{20, 50, 100\}$ for MP-SM; we use homology in dimensions 0 and 1 (except for the Cleves–Jain data, for which we use only dimension 0) with coefficients in the field $\Bbbk = \mathbb{Z}/11\mathbb{Z}$, which we just concatenate for MP-C, or combine linearly with coefficients chosen in $\{1, 5, 10\}$ for MP-SW. In general, the $n$ parameters of a multiparameter persistence module $M : \mathbb{R}^n \longrightarrow \mathrm{vec}$ represent quantities expressed in incomparable units, and are thus a priori incomparable; in order to account for this, we rescale each direction of the persistence module as follows: for MP-SW we choose a constant $c$ in $\{0.5, 1, 1.5\}$ for each direction, and scale by $c/\sigma$, with $\sigma \in \{0.001, 0.01, 1, 10, 100, 1000\}$; for MP-SM we choose a constant $c$ in $\{0.01, 0.125, 0.25, 0.375, 0.5\}$ for each direction.

### C.2 One-parameter experiments

We use an alpha filtration [35, Section 2.3.1] for the UCR data and, for the graph data, we filter graphs by the node degrees. For UCR data we use 0 and 1 dimensional homology; we take the sum the kernels evaluated on 0 and 1 dimensional features for the kernel methods and otherwise concatenate the features for the other vectorizations. For graph data we use extended persistence as in, e.g., [21].

The parameters of all vectorization methods are chosen by 10-fold cross validation. As classifier, we use a support vector machine with parameter $\gamma \in \{0.01, 0.1, 1, 10, 100\}$ and an RBF kernel, except for the kernel methods for which we use a precomputed kernel. The regularization parameter is chosen in $\{0.01, 1, 10, 100, 1000\}$. For sliced Wasserstein kernel (SWK) [21] we use $\sigma \in \{0.001, 0.01, 1, 10, 100, 1000\}$; for persistence images (PI) [1] we use the kernel bandwidth in $\{0.01, 0.1, 1, 10, 100\}$ and a resolution in $\{20, 30\}$; for persistence landscape (PL) [13] we let the number of landscapes to be in $\{3, 4, 5, 6, 7, 8\}$, and the resolution in $\{50, 100, 200, 300\}$; and for pervec (PV) we use a histogram with number of bins in $\{100, 200, 300\}$.

The reported accuracy is averaged over 10 train/test splits in the case of graphs; for UCR we use the given train/test split.

### C.3 Further graph experiments

We compare our methods to the multiparameter persistence methods based on the Euler characteristic ECP, RT, and HTn of [42]. We also compare against the state-of-the-art graph classification methods RetGK [81], FGSD [75], and GIN [80]. We choose these because they are the ones that performed the best in the analysis of [42]; in particular, we use the accuracies reported there. Parameters are as in Appendix C.1, except that we compute accuracy with 10-fold train/test split. The methods RetGK, FGSD, GIN and those from [42] also use 10-fold train/test splits for accuracy. Note that, for simplicity, and as opposed to [42], we do not cross-validate different choices of filtration, and instead we use the following three filtrations: the degree of the nodes normalized by the number of nodes in the graph, the closeness centrality, and the heat kernel signature [69] with time parameter 10. We believe it is possible that better scores can be obtained by considering various other combinations of filtrations. We also remark that choosing three or more filtrations is usually not possible with other persistence based methods ([42] being a notable exception).

All scores can be found in Table 5. One can see that our methods are competitive with the other topological baselines, which shows the benefits of using signed measure and barcode decompositions over raw Euler functions. It is also worth noting that topological methods achieve scores that are comparable with state-of-the-art, non-topological baselines.

### C.4 Pointcloud classification filtering Rips by density

In order to check that the performance of the sliced Wasserstein kernel in Section 4.2 can be improved by using a more robust filtration, as explained there, we use here a function-Rips filtration (Example 2) with a Gaussian kernel density estimate with bandwidth in $\{0.001 \cdot r, 0.01 \cdot r, 0.1 \cdot r, 0.2 \cdot r, 0.3 \cdot r\}$,

| Dataset | RetGK | FGSD | GIN | ECP | RT | HT nD | HSM-MP-SW | ESM-MP-SW | HSM-MP-C | ESM-MP-C |
|---|---|---|---|---|---|---|---|---|---|---|
| COX2 | **81.4(0.6)** | - | - | 80.3(0.4) | 79.7(0.4) | 80.6(0.4) | 77.9(1.3) | 78.2(0.8) | 78.1(2.4) | 78.4(1.8) |
| DHFR | 81.5(0.9) | - | - | 82.0(0.4) | 81.3(0.4) | 83.1(0.5) | 82.8(5) | **83.7(5.7)** | 81.5(3.1) | 81.3(3.2) |
| IMDB-B | 71.9(1.0) | 73.6 | **75.1(5.1)** | 73.3(0.4) | 74.0(0.5) | 74.7(0.5) | 74.7(5) | 74(3.9) | 72.9(2.9) | 74.2(3.9) |
| IMDB-M | 47.7(0.3) | **52.4** | 52.3(2.8) | 48.7(0.4) | 50.2(0.4) | 49.9(0.4) | 50.3(3.5) | 50.6(3.5) | 48.5(4.2) | 50.8(3.8) |
| MUTAG | 90.3(1.1) | **92.1** | 90(8.8) | 90.0(0.8) | 87.3(0.6) | 89.4(0.7) | 86.8(7.1) | 87.3(9.1) | 85.1(9) | 87.3(9) |
| PROTEINS | **78.0(0.3)** | 73.4 | 76.2(2.6) | 75.0(0.3) | 75.4(0.4) | 75.4(0.4) | 74.1(2) | 73.6(2.3) | 71.4(4) | 70(3.5) |

Table 5: Accuracy scores (averaged over 10-fold train/test splits) on graph datasets. Bold indicates best accuracy and underline indicates best accuracy among topological methods. For our methods and GIN, we report *standard deviation*, while RetGK reports *standard error*.

| Dataset | num. point clouds (train / test) | num. simplices per point cloud | HSM | MP-SW | MP-C |
|---|---|---|---|---|---|
| DistalPhalanxOutlineAgeGroup | 139 / 400 | 76153 | 7.69 | 24.87 | 4.50 |
| DistalPhalanxOutlineCorrect | 276 / 600 | 76153 | 11.61 | 56.25 | 6.64 |
| DistalPhalanxTW | 139 / 400 | 76153 | 7.78 | 24.39 | 4.75 |
| ProximalPhalanxOutlineAgeGroup | 400 / 205 | 76153 | 8.12 | 25.67 | 5.43 |
| ProximalPhalanxOutlineCorrect | 600 / 291 | 76153 | 12.39 | 56.83 | 7.77 |
| ProximalPhalanxTW | 205 / 400 | 76153 | 8.20 | 25.25 | 5.35 |
| ECG200 | 100 / 100 | 134137 | 2.66 | 1.76 | 1.22 |
| ItalyPowerDemand | 67 / 1029 | 1561 | 0.32 | 0.49 | 0.31 |
| MedicalImages | 381 / 760 | 147536 | 9.67 | 18.96 | 2.49 |
| Plane | 105 / 105 | 467321 | 4.52 | 3.03 | 2.42 |
| SwedishLeaf | 500 / 625 | 325625 | 18.34 | 53.83 | 8.81 |
| GunPoint | 50 / 150 | 529543 | 2.44 | 0.47 | 0.52 |
| GunPointAgeSpan | 135 / 316 | 529543 | 5.58 | 2.53 | 0.06 |
| GunPointMaleVersusFemale | 135 / 316 | 529543 | 5.56 | 2.53 | 0.05 |
| GunPointOldVersusYoung | 136 / 315 | 529543 | 5.60 | 2.50 | 0.06 |
| PowerCons | 180 / 180 | 467321 | 7.91 | 6.05 | 2.16 |
| SyntheticControl | 300 / 300 | 30913 | 4.54 | 13.33 | 3.82 |

Table 6: Time (in seconds) to go from point clouds to the Hilbert decomposition signed measure (column HSM), as well as time (in seconds) to go from the signed measure to the output of our proposed vectorizations (columns HSM-MP-SW and HSM-MP-C).

| Dataset | MP-K | MP-L | MP-I | HSM-MP-SW | HSM-MP-C |
|---|---|---|---|---|---|
| DistalPhalanxOutlineAgeGroup | 9227.1 | 1038.9 | 217.1 | 32.56 | 12.20 |
| DistalPhalanxOutlineCorrect | 36734.6 | 3492.6 | 833.7 | 67.87 | 18.25 |
| DistalPhalanxTW | 9396.4 | 577.7 | 138.4 | 32.18 | 12.53 |
| ProximalPhalanxOutlineAgeGroup | 11573.1 | 759.5 | 244.5 | 33.80 | 13.56 |
| ProximalPhalanxOutlineCorrect | 30822.7 | 2169.5 | 497.6 | 69.22 | 20.16 |
| ProximalPhalanxTW | 11641.7 | 375.4 | 93.4 | 33.45 | 13.56 |
| ECG200 | 1615.3 | 1355.6 | 269.0 | 4.42 | 3.88 |
| ItalyPowerDemand | 41918.1 | 1939.0 | 417.5 | 0.82 | 0.64 |
| MedicalImages | 147668.1 | 2404.7 | 599.5 | 28.63 | 12.16 |
| Plane | 2036.0 | 1065.0 | 249.2 | 7.55 | 6.94 |
| SwedishLeaf | 38045.7 | 3329.3 | 693.5 | 72.17 | 27.15 |
| GunPoint | 1977.0 | 1685.7 | 422.1 | 2.91 | 2.96 |
| GunPointAgeSpan | 14013.9 | 3945.6 | 1078.6 | 8.11 | 5.64 |
| GunPointMaleVersusFemale | 14069.9 | 4058.8 | 1097.0 | 8.10 | 5.62 |
| GunPointOldVersusYoung | 16668.1 | 5400.9 | 1388.5 | 8.10 | 5.67 |
| PowerCons | 8808.3 | 3234.8 | 811.4 | 13.97 | 10.08 |
| SyntheticControl | 13340.0 | 595.1 | 161.9 | 17.88 | 8.37 |

Table 7: Time (in seconds) taken by different vectorization methods for multifiltered simplicial complexes.

| Dataset | num. graphs | avg(std) simplices per graph | HSM | MP-SW | MP-C |
|---|---|---|---|---|---|
| COX2 | 467 | 84.6(8.2) | 0.91 | 6.21 | 2.24 |
| DHFR | 756 | 86.9(18.2) | 1.55 | 16.85 | 4.09 |
| IMDB-BINARY | 1000 | 116.3(113.3) | 81.49 | 14.28 | 3.97 |
| IMDB-MULTI | 1500 | 78.9(117.9) | 91.55 | 29.74 | 3.33 |
| MUTAG | 188 | 37.7(10.2) | 0.27 | 0.74 | 0.52 |
| PROTEINS | 1113 | 111.8(129.8) | 8.57 | 26.60 | 11.96 |

Table 8: Time (in seconds) to go from graphs to the Hilbert decomposition signed measure (column HSM), as well as time (in seconds) to go from the signed measure to the output of our proposed vectorizations (columns HSM-MP-SW and HSM-MP-C).

where $r$ is the radius of the dataset, chosen by cv. As one can see from the results in Table 9, MP-HSM-SW is indeed quite effective with this choice.

| Dataset | B1 | B2 | B3 | MP-K | MP-L | MP-I | MP-HSM-SW |
|---|---|---|---|---|---|---|---|
| DistalPhalanxOutlineAgeGroup | 62.6 | 62.6 | **77.0** | 67.6 | 70.5 | 71.9 | 71.9 |
| DistalPhalanxOutlineCorrect | 71.7 | 72.5 | 71.7 | 74.6 | 69.6 | 71.7 | **75.4** |
| DistalPhalanxTW | 63.3 | 63.3 | 59.0 | 61.2 | 56.1 | 61.9 | **66.9** |
| ProximalPhalanxOutlineAgeGroup | 78.5 | 78.5 | 80.5 | 78.0 | 78.5 | 81.0 | 84.4 |
| ProximalPhalanxOutlineCorrect | 80.8 | 79.0 | 78.4 | 78.7 | 78.7 | 81.8 | 84.5 |
| ProximalPhalanxTW | 70.7 | 75.6 | 75.6 | **79.5** | 73.2 | 76.1 | 78.0 |
| ECG200 | **88.0** | **88.0** | 77.0 | 77.0 | 74.0 | 83.0 | 87.0 |
| ItalyPowerDemand | 95.5 | 95.5 | 95.0 | 80.7 | 78.6 | 79.8 | 83.1 |
| MedicalImages | 68.4 | **74.7** | 73.7 | 55.4 | 55.7 | 60.0 | 67.9 |
| Plane | 96.2 | **100.0** | **100.0** | 92.4 | 84.8 | 97.1 | 99.0 |
| SwedishLeaf | 78.9 | 84.6 | 79.2 | 78.2 | 64.6 | 83.8 | 88.6 |
| GunPoint | 91.3 | 91.3 | 90.7 | 88.7 | 94.0 | 90.7 | 97.3 |
| GunPointAgeSpan | 89.9 | 96.5 | 91.8 | 93.0 | 85.1 | 90.5 | 97.8 |
| GunPointMaleVersusFemale | 97.5 | 97.5 | **99.7** | 96.8 | 88.3 | 95.9 | 98.4 |
| GunPointOldVersusYoung | 95.2 | 96.5 | 83.8 | 99.0 | 97.1 | **100.0** | 99.7 |
| PowerCons | **93.3** | 92.2 | 87.8 | 85.6 | 84.4 | 86.7 | 91.7 |
| SyntheticControl | 88.0 | 98.3 | **99.3** | 50.7 | 60.3 | 60.0 | 66.3 |

Table 9: Accuracy scores of baselines and multiparameter persistence methods on time series datasets. Boldface indicates best accuracy and underline indicates best accuracy among topological methods.

## C.5 The enrichment factor

A common approach for quantifying the performance of virtual screening methods [67] uses the *enrichment factor* $EF$, defined as follows. Given a query $q$ and a test of ligands $L$, with each ligand labeled as either *active* or a *decoy* with respect to the query, the virtual screening method is run on $(q, L)$ producing a linear order $O(L)$. Given $\alpha \in (0, 100)$,

$$EF_\alpha(q, L) := \frac{\left|\text{active molecules in first } (\alpha/100) \times |L| \text{ elements of } O(L)\right| \, / \, \left((\alpha/100) \times |L|\right)}{\left|\text{active molecules in } L\right| \, / \, |L|}.$$

We use the Cleves–Jain dataset [28], and follow the methodology of [67]. The dataset consists of a common set of 850 decoys $D$, and, for each of 22 targets $x \in \{a, \ldots, v\}$, two to three compounds $\{q_i^x\}$ and a set of 4 to 30 actives $L_x$. To quantify the performance of the method on the target $x$, one averages $EF_\alpha(q_i^x, L_x \cup D)$ over the compounds $\{q_i^x\}$. The overall performance, which is what is reported in Table 4 of the main article for different choices of $\alpha$, is computed by averaging these quantities over the 22 targets.

For topology-based methods for virtual screening which use the EF to assess performance, see [47, 16, 34].

## D Runtime experiments

We run these experiments in a computer with a Ryzen 4800 CPU, and with 16GB of RAM.

### D.1 Runtime of computation of Hilbert decomposition signed measure

**Hilbert function by reducing multiparameter to one-parameter persistence.** Let $(S, f : S \longrightarrow \mathbb{R}^n)$ be a filtered simplicial complex, and let $i \in \mathbb{N}$. Suppose we want to compute the Hilbert function of the homology persistence module $H_i(f) : \mathbb{R}^n \longrightarrow \text{vec}$ restricted to a grid, which, without loss of generality, we may assume to be $\{0, \ldots, m-1\}^n$ for some $m \in \mathbb{N}$. Given $a \in \{0, \ldots, m-1\}^{n-1}$ and $b \in \{0, \ldots, m-1\}$, denote $(a; b) = (a_1, \ldots, a_{n-1}, b)$. In particular, given $a \in \{0, \ldots, m-1\}^{n-1}$, we get a 1-parameter persistence module indexed by $b \in \{0, \ldots, m-1\}$ by mapping $b$ to $H_i(f)(a; b) \in \text{vec}$; we denote this persistence module by $H_i^a(f) : \{0, \ldots, m-1\} \longrightarrow \text{vec}$.

We proceed as follows. For each $a \in \{0, \ldots, m-1\}^{n-1}$, we use the one-parameter persistence algorithm [38] to compute the Hilbert function of the module $H_i^a(f)$. Thus, we perform $m^{n-1}$ runs of the one-parameter persistence algorithm. The worst case complexity of the one-parameter

persistence algorithm is $O\left((|S_{i-1}| + |S_i| + |S_{i+1}|)^3\right)$, where $|S_k|$ is the number of $k$-dimensional simplices of the simplicial complex, but it is known to be almost linear in practice [5].

In the UCR examples, since we are dealing with Vietoris–Rips complexes which can have lots of simplices, we rely on the edge-collapse optimization of [82] to speed up computations. Since the point clouds are not too large, we do not perform any further optimizations, but we mention that optimizations like the ones of [14, 66] are available for dealing with large point clouds.

**Runtimes.** In the fourth column (HSM) of Tables 6 and 8, we report the time taken for computing the Hilbert decomposition signed measure with resolution $k = 200$ starting from a filtered simplicial complex. See the experiments section in the article for a description of these filtered simplicial complexes. Then, in the last two columns of Tables 6 and 8 (MP-SW and MP-C), we report the time taken for computing our proposed vectorizations starting from the Hilbert decomposition signed measure.

One can see that, in both tables, the bottleneck is usually either the HSM or the MP-SW computation, as MP-C is quite fast to compute. Overall, the whole pipeline (decomposition + vectorization) can be achieved in a quite reasonable amount of time, as the running times never go beyond $10^2$ seconds, which is quite efficient in the realm of topological methods (see also next section).

### D.2 Runtime of whole pipeline

In Table 7, we compare the runtime of our full pipeline (from the point clouds obtained from the UCR datasets to the output of our vectorizations) to that of other pipelines based on multiparameter persistence. For other pipelines, we use the numbers in [20].

It is quite clear that our pipeline is much faster than the other topological baselines, by several orders of magnitude. This is generally due to the fact that Hilbert decomposition signed measures can be computed in the same amount of time than fibered barcodes (which are needed by the baselines), and can be turned into vectors in a single step with one of our proposed vectorizations, while other baselines require vectorizing all elements of the fibered barcodes.

## E  Stability experiments

In this experiment, we test our main stability results, Theorems 1 to 3. We fix a simplicial complex $K$ and consider filtrations $f : K \longrightarrow \mathbb{R}^2$ using the lower-star filtration associated to functions $K_0 \longrightarrow \mathbb{R}^2$ on the vertices of $K$. We construct functions on the vertices of $K$ as follows: we treat the function as a vector of dimension $|K_0|$, start with a constant vector, and iteratively add uniform random noise to each component. This is effectively a random walk that, at each step, produces a function filtering $K$. Thus, each random walk produces a set of filtering functions $\{f_i\}_{1 \le i \le k}$.

For each random walk (shown with a different color), we consider the $L^1$-distances between functions $\|f_i - f_j\|_1$, the Kantorovich–Rubinstein distances between Hilbert signed measures $\|\mu_{H_0(f_i)} - \mu_{H_0(f_j)}\|_2^{\mathsf{KR}}$, the sliced Wasserstein kernel distances between vectorizations $\|\mathsf{HSM\text{-}SW}(\mu_{H_0(f_i)}) - \mathsf{HSM\text{-}SW}(\mu_{H_0(f_j)})\|_{\mathcal{H}}$, and the $L^2$-distances between convolution vectorizations $\|\mathsf{HSM\text{-}C}(\mu_{H_0(f_i)}) - \mathsf{HSM\text{-}C}(\mu_{H_0(f_j)})\|_2$. See Fig. 4.

The fact that the points in the plot lie below a line with positive slope passing through the origin is a consequence of our stability results for the Hilbert signed measure and for the vectorizations. The fact that the points in the plot lie above a line passing through the origin (at least for points with sufficiently small $x$-coordinate), shows that, for this kind of data, our proposed vectorizations are a strong invariant, meaning that it is able to distinguish different filtering functions.

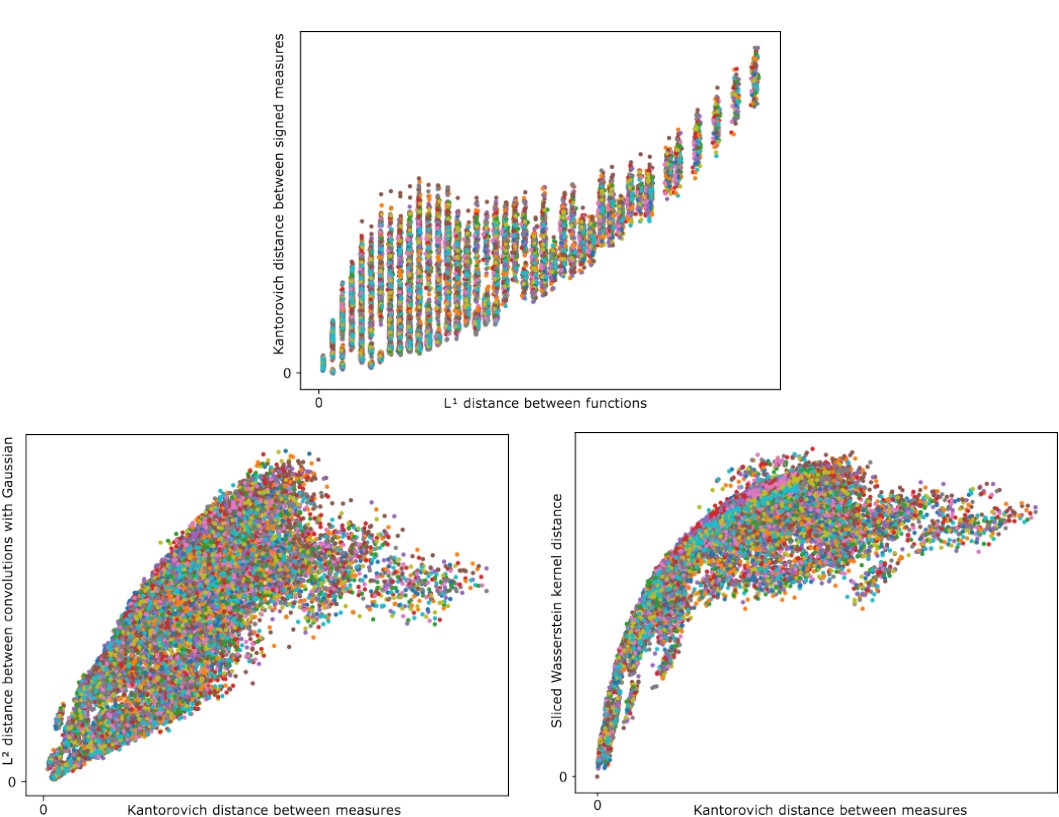

Figure 4: *Top:* $L^1$-distance between filtering functions and Kantorovich–Rubinstein distance between the associated Hilbert signed measures. *Bottom:* Kantorovich–Rubistein distance between signed measures and distance between the vectors produced by our vectorizations. Different colors indicate different runs of the random walk used to construct the filtering functions. Axis do not have scale since scale depends on the choice of norms and of vectorization parameters.

