# OpenReview forum: "Stable Vectorization of Multiparameter Persistent Homology using Signed Barcodes as Measures"
_NeurIPS.cc/2023/Conference — NeurIPS 2023 poster_

### Official Review · Reviewer_csyn · 2023-06-25

**Soundness:** 4 excellent
**Presentation:** 3 good
**Contribution:** 4 excellent
**Rating:** 8
**Confidence:** 5

**Summary:**

In this paper, the authors are addressing a very critical need in topological data analysis (TDA), vectorization of multiparameter persistence (MPH). Persistent homology (PH) is the key method in TDA, but in its current form, it allows only a single function to use in its key process, filtration. By enabling multiple functions, MPH is a natural generalization of (PH) with much finer information, however, there are several mathematical obstructions to use them effectively in applications.

In this paper, the authors propose an effective way to vectorize the barcode information in the MP module by sacrificing some to keep the computation feasible and process practical. They offer two versions of MP vectorizations where one being kernel and the other being direct vectorization, both can be effective in different settings. The authors made extensive experiments in several settings, including point cloud, graph classification, and virtual screening to compare their model with other MP vectorizations and SOTA models. Their model consistently outperforms existing MP vectorizations.



**Strengths:**

The authors use a recent idea of signed barcodes cleverly to obtain practical vectorizations. There is a significant need to employ MPH idea with good and feasible vectorizations, and this paper makes a valuable contribution in this direction.

They propose two versions of their vectorizations, a kernel (MP-SW) and a direct vectorization (MP-C). In the past years, depending on the domain, we see that one can perform better than the other from our experience with the applications of single persistence. So, this versatility of outputs is very valuable for ML applications.

Computational time is significantly better than existing MP vectorizations. Combining with its good performance, this might be the most important contribution of the paper from ML perspective.

Their extensive experiments in various settings show that their model consistently outperforms the existing MPH vectorizations.

**Weaknesses:**

The main weakness is that the paper might be too technical for non-experts and ML audience. However, considering the depth of the problem, I can see that the authors did an enormous effort to make this important subject accessible to a wider audience.

While new vectorizations consistently outperform the existing MP models in point cloud settings, the performance is not as strong in graph classification and specifically, virtual screening cases (TODD also uses a version of MPH).

As MPH is an involved process, hyperparameter tuning might need serious expertise in TDA in real life applications.



**Questions:**

This is more of a question/remark. A low-hanging fruit might be to combine (concatenate) Hilbert functions (MP-H) with MP-HSM-C (or MP-ESM-C) as you already compute MP-H during the process if I understand the process right. Considering the closeness of the performances in graph classification and virtual screening, this combined vectorization might perform better.

**Limitations:**

As mentioned in weaknesses, the main limitations are scalability and hyperparameter tuning which could be very tricky, especially choosing the right grid size. Hence, performance vs. computational feasibility can be a problem in large datasets.

---

> ### Author Rebuttal · Authors · 2023-08-09
>
> Thank you for your time and feedback.
>
> - *Performance on graph data.*
> We agree that the results in section 4.3 (graph data) are not as good as those in section 4.2 (point cloud data).
> We note, however, that the performance on the virtual screening task of our method is very good; ToDD is a supervised method, whereas in this task we are using our vectorizations in a completely unsupervised way (we included ToDD to give an idea of how well TDA-based tools can work on this type of data).
> With respect to section 4.3, we are using an invariant (the Hilbert function or, equivalently, the Hilbert signed measure) that is strictly weaker than that used by other topological methods (such as the rank invariant), which could be the reason.
> In particular, we believe that the issue is not with the vectorizations themselves, and that the gap in performance could be bridged by using signed barcodes/measures coming from the rank invariant.
>
> - *Combine/concatenate MP-H and MP-HSM-C (or MP-ESM-C) to get better performance.*
> We think this is an interesting idea which should be experimented with.
> In this paper, we have decided to keep the approach as straightforward as possible to try to isolate the performance of the vectorization from the performance of the machine learning model and other application specific choices.
> We address the tuning of the approach, including combination of different vectorizations, in ongoing work.

---

> > ### Comment · Reviewer_csyn · 2023-08-18
> >
> > Thank you very much for your detailed answers. I have no further questions. Good luck with your submission.

---

### Official Review · Reviewer_VRYZ · 2023-07-06

**Soundness:** 3 good
**Presentation:** 2 fair
**Contribution:** 2 fair
**Rating:** 4
**Confidence:** 4

**Summary:**

The paper vectorizes data descriptors coming from multiparameter persistent homology for classifying point clouds and measuring similarity between graphs extracted from databases of times series and molecules.

**Strengths:**

The paper includes rigorous definitions and proves (in the appendices) three theorems from section 4.

**Weaknesses:**

The paper seems to over-advertise the strengths of persistence.

Lines 1-2 claim that "Persistent homology (PH) provides topological descriptors for geometric data, such as weighted graphs". If the authors really meant a topological classification of graphs, this classification has been known to Euler in the 18th century, so there is no need to re-invent topological invariants of graphs by using persistence.

Lines 22-23 claim that "TDA methods usually require the sole knowledge of a metric or dissimilarity measure on the data". In fact, the definition already of persistence requires a choice of filtration of simplicial complexes on given data points, which seriously affects the resulting persistence whose further analysis requires many more extra parameters, see lines 351-352: " Our pipelines, including the choice of hyperparameters for our vectorizations, rely on the cross validation of several parameters, which limits the number of possible choices"

Examples 1-3 are definitions or comments, not illustrative examples that could help the readability.

The questions and limitations below include further concerns about experiments and comparisons with past work.

**Questions:**

Lines 14-15 claim that "The resulting feature vectors are easy to define and to compute". If this is true, could the authors write a full easy definition of these vectors in a few lines?

Concerning the computation, line 245-256 seem more honest: "We discretize the computation of the sliced Wasserstein kernel by choosing directions {θ1, . . . , θd} ⊆ Sn−1 uniformly at random". Are there any theoretical guarantees for this approximate computation and what is the asymptotic cost in the size of the input?

The authors correctly highlight the importance of stability (Lipschitz continuity) under noise. However, the practical experiments in section 4.2 use discrete grids, which makes all features discontinuous under perturbations of times series or point clouds. Is it possible to avoid this discretization?

Section 4.3 discusses datasets of graphs coming from social networks, biology, and medicine. How can edges be justified in these experimental graphs?

In social networks, friendships or other links between people have different strengths and are always subjective. In molecules, even the strongest covalent bonds between atoms have no strict definitions and only abstractly represent inter-atomic interactions, so there are physical sticks or strings between them. That is why most databases of molecules and materials such as QM9 contain only atomic positions but no links between them.

Which of the invariants in Appendix B are provably Lipschitz continuous under perturbations of points, sat in the bottleneck distance on the persistence diagram?

Which of these invariants are faster or stronger than the isometry invariants from Widdowson et al (CVPR 2023)?

Which (dis)similarity functions mentioned in the paper satisfy all metric axioms? The importance of metric axioms including the triangle inequality was theoretically justified in the paper arXiv:2211.03674, which should be cited in every work on clustering and metric distances because any clustering based on non-metrics is not trustworthy.

**Limitations:**

The main limitation of the 1-parameter persistence is its weakness as an isometry invariant, which should have been clear to all experts in computational geometry many years ago but was demonstrated only recently. The paper by Smith et al (arxiv:2202.00577) explicitly constructs generic families of point clouds in Euclidean and metric spaces that are indistinguishable by persistence and even have empty persistence in dimension 1. These examples can be easily extended to more than one parameter at least for some filtrations.

Though Topological Data Analysis was largely developed by mathematicians, the huge effort over many years was invested into speeding up computations, rather surprisingly, instead of trying to understand the strengths and weaknesses of persistent homology, especially in comparison with the much simpler, faster, and stronger invariants of clouds under isometry.

Persistence in dimension 0 was extended to a strictly stronger invariant mergegram by Elkin et al in MFCS 2020 and Mathematics 2021, which has the same asymptotic time as the classical 0D persistence and is also stable under perturbations of points.

A SoCG 2022 workshop included a frank discussion concluding that there was no high-level problem that persistent homology solves. Is there such an ultimate problem for multi-parameter persistence? In fact, persistence as an isometry invariant essentially tries to distinguish clouds of unlabeled points up to isometry, not up to continuous deformations since even non-uniform scaling changes persistence in a non-controllable way.

On the other hand, the isometry classification problem for point clouds was nearly solved by Boutin and Kemper (2004), who proved that the total distribution of pairwise distances is a generically complete invariant of point clouds in any Euclidean space. The remaining singular cases were recently covered by Widdowson et al in NeurIPS 2022 and CVPR 2023.

---

> ### Author Rebuttal · Authors · 2023-08-09
>
> Thank you for your time and feedback.
>
> We start with two clarifications: Our applications go well beyond classification of point clouds up to isometry, and even in the point cloud application, we do not seek a strong isometry invariant since such invariants are necessarily highly sensitive to, e.g., outliers.
> It is also beyond the scope of this rebuttal to discuss the efficacy of PH, which is well-established given the vast amount of existing applications in ML ([Hensel, Moor, Rieck, 2021] and references).
>
> - *"Lines 1-2 [...] using persistence"* We do not understand this comment: the quoted sentence does not mention "topological classification of graphs". All we are saying here is that persistence allows for the use of topology to produce descriptors of geometric data. A weighted graph can be taken as a single graph, but persistence allows one to incorporate much richer information into topological descriptors by using filtrations derived from the weights.
>
> - *"Lines 22-23 [...] possible choices"* The fact that our method has hyperparameters does not prevent it from being applicable in arbitrary metric spaces or weighted graphs. One can resort to default choices as in Section 4.4.
>
> - *"Examples 1-3 [...] readability"* Ex 1 provides a classical example of 1-dimensional simplicial complex; Ex 2 describes a classical family of multifiltered complexes; Ex 3 gives a usual way of getting a persistence module from a filtration. We do not see why these examples should not be helpful to the reader.
> Does the reviewer have other examples in mind?
>
> - *"Could the authors [...] few lines"* The Hilbert function of a module is its pointwise dimension (def 2), and the Hilbert signed measure is computed by Mobius inversion of the Hilbert function (amounting to a convolution with a kernel of small support; this is reference to [41, Rmk. 3] in line 213). The formulas for vectorizations are explicitly given in defs 6 and 7; and the computation is described in lines 232 and 245, respectively. We will be happy to make this comment into pseudocode, if the reviewer believes this will improve readability.
>
> - *"Are there any [...] input"* There is no approximation: our definition of the Wasserstein kernel takes any measure on the $(n-1)$-sphere, and our stability result holds for any such measure.
> This allows us to use a discrete measure so that the integral can be evaluated exactly. One could consider the approximation of SWK with uniform measure on the sphere by a SWK with discrete measures, as in [Carriere et al. 2017] and the analysis would be almost identical. For space concerns, we decided to have the theory of the paper focus on stability results and not on injectivity/inverse results.
> The time complexity of the computation of the SWK is linear in the number of point masses of the discrete measure on the $(n-1)$-sphere.
>
> - *"Is it possible to avoid this discretization?"* Discretizations make the vectors discontinuous, but, crucially, Lipschitz stability theorems still hold up to the discretization size (on grids of step size $\epsilon$, stability holds up to additive term of $O(\epsilon)$), so near-by datasets still get near-by vectors, and approximation can be controlled by the user.
> It is possible to avoid discretizations: the computation of the Hilbert and Euler signed measures can be done exactly since fp MP modules can be described exactly over some data-dependent finite grid.
> So the SWK can be computed exactly on any finite point measure.
> The convolution vectorization does require a grid if one seeks a vectorization (i.e., a Euclidean vector for each sample); however, if a kernel method is sufficient, one can dispense with discretizations because the $L^2$ inner product between Gaussian mixtures can be computed exactly (see, e.g., the vectorization of [Reininghaus et al. 2015]).
>
> - *"How can edges [...] graphs"* We seek to understand the performance of our vectorizations for graph classification, so we take the input datasets at face value like previous papers.
> While interesting, the discussion about the validity of edgers is outside the scope of the paper.
>
> - *"Which of [...] persistence diagram?"* We are not sure to understand this question: since the invariants being referred to are invariants of multiparameter persistence modules, there is, a priori, no persistence diagram. Could you please clarify this question?
>
> - *"Which of these [...] (CVPR 2023)?"* In Section 4.2, we do not seek to use strong isometric invariants for point clouds such as the ones in (CVPR 2023) because we want stability under, e.g., outliers.
> We rather see the point clouds as empirical measures, and use density-geometric bifiltrations whose one-parameter slices are stable under Wasserstein perturbations [Chazal, Cohen-Steiner, Mérigot (2011)].
>
> - *"Which (dis)similarity [...] axioms?"* Our vectorizations have the space of finite signed measures with total mass zero as domain, equipped with the Kantorovitch norm, which is an honest norm; thus, its induced metric satisfies all metric axioms. The codomain of our vectorizations are Hilbert spaces, where the metric is derived from the inner product and therefore is an honest metric.
>
> - *"About an ultimate problem/goal for multiparameter persistence"* Besides recent applications of multiparameter persistence (see motivating paragraph for MPH, for now in the response to reviewer 61ca), various theoretical papers show that TDA with several parameters/directions can address difficult practical problems: [Blumberg, Lesnick. Found Comp Math, 2022] shows that two-parameter persistence provides Gromov-Prokhorov Lischitz-stable invariants (i.e., robust to outliers) of metric probability spaces that do not require user chosen parameters, such as a KDE bandwidth. [Curry, Mukherjee, Turner. Trans Amer Math Soc Ser B, 2022] shows that several one-parameter filtrations, taken together, can uniquely characterize simplicial complexes in Euclidean space up to action of the orthogonal group.

---

> > ### Comment · Reviewer_VRYZ · 2023-08-12
> > **remaining questions**
> >
> > Thank you for the reply.
> >
> > >efficacy of PH, which is well-established given the vast amount of existing applications in ML ([Hensel, Moor, Rieck, 2021] and references)
> >
> > The strong claim about efficacy requires serious justifications not by including a vast amount of papers but by quoting rigorously proved theorems. The widely cited stability of persistence gives only an upper bound under perturbations. Such a Lipschitz upper bound is straightforward for many easier and faster invariants, for example, the total distribution of pairwise distances.
> >
> > The key ignored difficulty was the lack (actually, impossibility) of a lower bound, because the persistence doesn't change or even remains trivial (fully empty) under all perturbations for a generic family of point clouds, even in the plane.
> >
> > While Theorems 1-3 prove upper bounds, can the left hand sides of these inequalities be zeros for different functions and measures?
> >
> > Could the authors please specify exact places, where the paper [Hensel, Moor, Rieck, 2021] discusses efficacy of PH? The Frontiers Media is actually known for the facts of controversial reviewing https://en.wikipedia.org/wiki/Frontiers_Media#Controversies
> >
> > > persistence allows one to incorporate much richer information into topological descriptors by using filtrations derived from the weights.
> >
> > How does persistence incorporate topological descriptors if the persistence changes even under non-uniform scaling, much worse under more flexible topological deformations?
> >
> > >The fact that our method has hyperparameters does not prevent it from being applicable in arbitrary metric spaces or weighted graphs. One can resort to default choices as in Section 4.4.
> >
> > What choices were called default here: "grid size of k = 1000 for all methods" in line 336? How is this number 1000 justified?
> >
> > >"Which of [...] persistence diagram?" We are not sure to understand this question: since the invariants being referred to are invariants of multiparameter persistence modules, there is, a priori, no persistence diagram.
> >
> > Which of the invariants in Appendix B are provably Lipschitz continuous under perturbations of given points?
> >
> > >[Curry, Mukherjee, Turner. Trans Amer Math Soc Ser B, 2022] shows that several one-parameter filtrations, taken together, can uniquely characterize simplicial complexes in Euclidean space up to action of the orthogonal group.
> >
> > Main Theorem 7.14 in this paper is proved only for complexes in general position. A reconstruction in general position was proved for the harder case of point clouds (without any edges or simplices) by Boutin and Kemper "On reconstructing n-point configurations from the distribution of distances or areas" in Advances in Applied Mathematics (2004) by using the much simpler distribution of pairwise distances without any slower persistence.
> >
> > Do the words "up to action of the orthogonal group" mean that the persistence-based objects from the paper above should be compared by rotations?
> >
> > To check if original complexes are rigidly equivalent, it's easier to fix their centers of mass at the origin and rotate given vertices, see Alt et al "Congruence, similarity, and symmetries of geometric objects" in Discrete and Computational Geometry (1988) and Brass et al "Testing congruence and symmetry for general 3-dimensional objects" in Computational Geometry (2004).

---

> > > ### Author Response · Authors · 2023-08-13
> > >
> > > We thank you again for the discussion.
> > >
> > > - *"The strong claim about efficacy requires [...]"*
> > > [Hensel, Moor, Rieck, 2021] is about the diversity of "existing applications in ML".
> > > Efficacy is not only about theoretical results but also about practical performances, especially in an ML venue such as this one. In this regard, a number of TDA works have been published every year at NeurIPS since 2019
> > > (e.g.,
> > > [Zhao, Wang. NeurIPS 2019],
> > > [Hu, Li, Samaras, Chen. NeurIPS 2019],
> > > [Kim, Kim, Zaheer, Kim, Chazal, Wasserman. NeurIPS 2020],
> > > [Carrière, Blumberg. NeurIPS 2020],
> > > [Vishwanath, Fukumizu, Kuriki, Sriperumbudur. NeurIPS 2020],
> > > [Chen, Coskunuzer, Gel. NeurIPS 2021],
> > > [Birdal, Lou, Guibas, Simsekli. NeurIPS 2021],
> > > [Zheng, Zhang, Wagner, Goswami, Chen. NeurIPS 2021],
> > > [Demir, Coskunuzer, Gel, Segovia-Dominguez, Chen, Kiziltan. NeurIPS 2022],
> > > [Akcora, Kantarcioglu, Gel, Coskunuzer. NeurIPS 2022],
> > > [Turkes, Montufar, Otter. NeurIPS 2022]).
> > >
> > > - *"While Theorems 1-3 prove upper bounds, can the left hand sides of these inequalities be zeros for different functions and measures?"*
> > > We believe that there may be lower bounds for theorem 2 for point measures with bounded number of masses, and for theorem 3 only locally and also for point measures with a bounded number of masses.
> > > A lower bound for theorem 1 is probably not possible in full generality.
> > > Yet, the experiments in our paper show that the invariants are still expressive enough in a variety of contexts.
> > >
> > > - *"How does persistence incorporate topological descriptors [...] deformations?"*
> > > Persistence is known to be a homeomorphism invariant of filtered topological spaces, in that the composite of a homeomorphism with a filtering function has the same persistence barcode as the filtering function.
> > > How this general invariance translates to transformations of the data depends on the (application dependent) construction that turns data into filtered spaces.
> > > In any case, we don't mean that persistence "incorporate topological descriptors" as you write, but rather that it turns a topological descriptor, such as homology, into a richer descriptor that encodes additional information captured through the filter function.
> > > Note also that multiparameter persistence is what naturally arises when considering multiple filter functions.
> > >
> > > - *"What choices were called default here [...]"*
> > > To avoid the parameter $k$, one can proceed as outlined in the response to one of your previous questions, by noticing that the distance between the vectors returned by our vectorizations has a closed form solution.
> > > To simplify the implementation, we simply chose a grid that is as fine as possible, given our computing resources.
> > >
> > > - *"Which of the invariants in Appendix B [...]"*
> > > The invariants in Appendix B are for persistence modules.
> > > How the persistence modules are constructed from the data, and thus, how they vary with respect to perturbations of the data, is application dependent.
> > > We note, however, that the papers that introduce some of the invariants of Appendix B (e.g., persistence landscape and its multiparameter versions, and multiparameter persistence kernel) prove stability results, usually in terms of the interleaving or matching distance between multiparameter persistence modules.
> > >
> > >
> > > - *"[Curry, Mukherjee, Turner. [...]"*
> > > We mentioned this paper as one particular example of a difficult practical problem that can be addressed using multiple parameters/directions.
> > > The references you provided, and the discussion about their relationship to the paper we mentioned, are very interesting.
> > > However, we believe this discussion is outside the scope of this rebuttal, since it is only tangentially related to our submission.
> > >
> > > Based on your feedback and the references you have provided so far, we propose to add a clarification in Section 4.2, explaining that, in that application, we are not trying to do point cloud classification up to isometry (because of the possibility of outliers), a problem for which well-developed methods exist as you have pointed out.

---

> > > > ### Comment · Reviewer_VRYZ · 2023-08-21
> > > > **Thank you for the reply**
> > > >
> > > > Thank you for the reply.
> > > >
> > > > In lines 295-296, the paper chose to "compare against non-topological, state-of-the-art baselines: Euclidean nearest neighbor (B1), dynamic time warping with optimized warping window width (B2), and constant window width (B3)".
> > > >
> > > > Since the word "neighbor" appears only once in the whole paper with appendices, would it be possible to clarify please if B1 has only one distance from every point to its first nearest neighbor in a given cloud?
> > > >
> > > > On what proportion of the datasets in Table 2 have the methods MP-... outperformed all the chosen baselines B1, B2, B3?
> > > >
> > > > Why wasn't the much easier and more powerful baseline of all pairwise distances used?
> > > >
> > > > The distribution of all pairwise distances between m given points is computable in time O(m^2), Lipschitz continuous under perturbations and was proved to be enough for reconstructing any generic Euclidean point cloud by Boutin and Kemper in Adv. Appl. Math. (2004).
> > > >
> > > > >a number of TDA works have been published every year at NeurIPS since 2019
> > > >
> > > > The last response contained even more citations without referring to specific results in these papers, though the request was to provide exact claims or page numbers. Has any of these papers compared persistent homology with the distribution of pairwise distances?
> > > >
> > > > At the same time, when another reference [Curry, Mukherjee, Turner...] was questioned, the authors "mentioned this paper as one particular example of a difficult practical problem that can be addressed using multiple parameters/directions" without even acknowledging the fact that this problem was solved not later than in 2004 by a much simpler invariant.
> > > >
> > > > >Efficacy is not only about theoretical results but also about practical performances
> > > >
> > > > The practical performance on a specific dataset is only an example, not a guarantee for other data beyond training sets.
> > > >
> > > > Here is a practical illustration. The prediction that any odd integer (greater than 2) is prime (has no divisors greater than 1) works well for 3,5,7, not 9 (an experimental error), 11, 13, hence achieving the accuracy of 5/6, more than 80%. One can easily improve the accuracy by forbidding multiples of 3 (greater than 3). Then the first counter-example will be 25=5*5 (or 35=5*7 if perfect squares are also excluded as obvious non-primes). One can easily continue forbidding multiples of 5, 7, and so on, making the accuracy nearly 100% on any finite data.
> > > >
> > > > Should we use such an algorithm for detecting primes in cryptography-based algorithms for financial transactions in the real world?
> > > >
> > > > No, because the infinite set of primes will always provide counter-examples. The modern world exists due to mathematically proved results, not due to unjustified perfomances. Examples prove nothing because (sometimes infinitely) many counter-examples can still exist.

---

### Official Review · Reviewer_cn3Z · 2023-07-07

**Soundness:** 3 good
**Presentation:** 4 excellent
**Contribution:** 3 good
**Rating:** 6
**Confidence:** 3

**Summary:**

The authors introduce first vectorizations of multiparameter persistent homology (MPH) via signed barcodes, that are easy to compute and shown to be stable. The two proposed vectorizations often outperform the state of the art MPH methods on a variety of data sets.

**Strengths:**

(S1) The paper is clearly organized and written, it reads really well.

(S2) The experiments are rather extensive and the results seem convincing.

**Weaknesses:**

I did not identify major weaknesses in this work, besides some confusion about the experimental settings addressed in question (Q2) below. I am open to raising my score if this issue is properly addressed.

**Questions:**

(Q1) The proposed approach is theoretically weaker than some other (Appendix A, Proposition 1), but it outperforms them. This should be at least briefly commented/discussed.

(Q2) It seems that the two and three filtrations used respectively in Sections 4.2 and 4.3 for the calculation of proposed MPH featurizations are not the same as the other considered MPH approaches from the literature? How is the comparison fair/reasonable then?

(Q3) Can you comment on why MP-HSM-SW is much better for some, whereas MP-ESM-C performs significantly better for other data sets?

Other minor comments:

-	Definition 6: Name the introduced notion K * \mu.
-	Line 81: Briefly summarize the conclusions of the runtime assessment, i.e. that your pipeline is much faster than the other topological baselines, by several orders of magnitude (Appendix D.2, Table 3).
-	Theorems and Propositions: For better readability, it would be useful to name the theoretical results.
-	Section 4.1: Describe the data and filtration.
-	Line 301: “As one can see from the results in Table 2, MP-HSM-SW is quite efficient”. Rephrase, since Table 2 does not show the computation times, but accuracy.
-	Table 4 caption: MP-HSM-C -> MP-ESM-C?
-	References: Be consistent between capital case vs. lower case for journal names, and also between full and shortened names, e.g. Journal of Machine Learning Research or J. Mach. Learn. Res. (similarly, Discrete Comput. Geom, Found. Comput. Math, …). Correct 2d, Xgboost, ucr, betti, Tudaset.


**Limitations:**

The limitations and future work are discussed in detail.

---

> ### Author Rebuttal · Authors · 2023-08-10
>
> Thank you for your time and feedback.
>
> - *Comment on/discuss the fact that the proposed approach is theoretically weaker than other approaches (Appendix A, Proposition 1), but it outperforms them.*
> We believe this is due to the fact that, despite being weaker than the rank invariant, using the Hilbert/Euler functions allows us to leverage and use the vectorization techniques that have worked the best for one-parameter persistence (in our case, Gaussian convolution and sliced Wasserstein kernel).
> In other words, while the rank invariant is richer, the ability to use it in machine learning pipelines is (currently, and to the best of our knowledge) limited in the sense that it forces one to use more complicated vectorizations.
> In the original submission, we have commented on this briefly in line 349 of the Conclusions.
> We agree that this is too succinct, and we will add the following more detailed comment:
>
> >[...] strictly weaker descriptors.
> >We believe that this is due to the fact that using the Hilbert and Euler signed measures allows us to leverage well-developed vectorization techniques shown to have good performance in one-parameter persistence.
> >We conclude from this that the way topological descriptors are encoded is as important as the discriminative power of the descriptors themselves.
>
> - *Filtrations of Sections 4.2 and 4.3 are not the same as the other considered MPH approaches from literature.*
> We agree and thank you for pointing this out.
> We have rerun the experiments of sections 4.2 and 4.3.
> For section 4.2, we now used alpha complex + DTM filtration as in [Carrière, Blumberg. NeurIPS 2022].
> We are including the new results for the first 6 UCR time series datasets of the original submission in Table 1 of the rebuttal PDF we submitted.
> For section 4.3, we now used Heat Kernel Signature (with time 10) + Ricci curvature as in [Carrière, Blumberg. NeurIPS 2022].
> We are including the new results for the graph datasets of the original submission in Table 2 of the rebuttal PDF we submitted.
> The takeaway for both tables is the same: our methods are outperforming previous MPH vectorization methods for these tasks.
> We note two interesting things: SWK is doing slightly worse than using Rips+KDE as in the original submission (but still better than previous MPH vectorizations), while the convolution-based vectorization is doing better; we believe that the improved performance of the convolution-based vectorization could be due to some implementation improvements having to do with how measures are snapped to a grid.
> If the paper is accepted, we will include the full table as well as the Hilbert and Euler function columns (not part of previous work, but included in the original submission for the sake of comparison), which are now missing due to time constraints.
>
> - *Comment on why MP-HSM-SW is much better for some, whereas MP-ESM-C performs better for other data sets.*
> Both vectorizations have advantages and drawbacks.
> The convolution-based vectorization requires the choice of a relatively small grid in order to make training practical, whereas SWK can take a much finer grid since it does not result in a more costly vectorization.
> But, the SWK can be sensitive to outliers in the form of far-away point masses, since transporting them can be costly in terms of Wasserstein distance; meanwhile, for the convolution-based vectorization a far-away point mass just results in a bump whose distance to zero is bounded regardless of how far it is.
> We believe that the best method depends highly on the data type.
>
> - *Other minor comments.*
> Thank you for spotting these; we will address them.
> One question: when you say "For better readability, it would be useful to name the theoretical results", do you mean giving a short description of the result like "Theorem 3 (Stability of sliced Wasserstein)"?

---

> > ### Comment · Reviewer_cn3Z · 2023-08-14
> >
> > Thank you for the clarifications. Some final comments:
> >
> > - Thanks for running the experiments for the filtrations considered in the literature. In the final version of the paper, I suggest to include both these filtrations, as well as your original choice, since I assume that there is a reason you opted for those. In case you include at least a brief discussion on the choice of filtrations and some insights on how they influence the results, and as suggested, I will increase my rating of the paper.
> >
> > - Indeed, you understood my comment about the naming of definitions and theoretical results, I think it helps the readability, but this is a matter of style so I leave the final decision to you.

---

> > > ### Author Response · Authors · 2023-08-16
> > > **Answer to reviewer cn3Z**
> > >
> > > Thank you for the suggestions, which we will follow.
> > >
> > > We definitely agree with you that using the exact same filtrations is the way to go for a fair comparison and these are the results we will present to make the point that, in these applications, our vectorizations lead to improved performance compared to previous methods. The set of results that appears in our original submission will also be included (and they will be rerun for the convolution vectorization due to the implementation improvements mentioned in our previous response to you). Here we will include a discussion comparing the different choices, saying in particular that the performance gain could be attributed to the better stability properties of Rips, in the case of the UCR data, and to the larger number of parameters, in the case of the graph data.
> > >
> > > Specifically, we will include the two sets of experiments (improved original + rebuttal) in the paper, as well as a discussion like the following about how the choices influence the results (which in particular contains the motivation for the original choice of filtrations).
> > >
> > > "Note that for the UCR data, previous work in the literature used bifiltrations obtained with Alpha and DTM. In addition to these, in this article we also run experiments with (a) Rips instead of Alpha because Rips scales better with the embedding dimension of the point cloud (and thus allows for better generalization of the application of our method to point clouds with outliers), and (b) with KDE instead of DTM because KDE behaves similarly to DTM while being easier to fine tune as it is more standard and has been more studied theoretically. Note also that for the graph data, we also decided to experiment with using more than two parameters since our methods apply directly in this case, in contrast to methods from the literature (such as multiparameter persistence images and GRIL)."
> > >
> > > We will be happy to discuss any further questions or comments.

---

### Official Review · Reviewer_61ca · 2023-07-08

**Soundness:** 3 good
**Presentation:** 2 fair
**Contribution:** 2 fair
**Rating:** 6
**Confidence:** 3

**Summary:**

This work promote the use of signed barcodes for feature generation, and proposed the feature generation pipeline based on the signed bar codes,
a. the work introduces two general vectorization techniques for signed barcodes;
b. the authors prove Lipschitz-continuity results that ensure the robustness of the proposed entire feature generation pipeline
c. the practical performance of the proposed pipeline is compared to other baselines in various supervised and unsupervised learning tasks.

All of my questions are fully addressed by the rebuttal. I am satisfied with the explanations. I think the work is solid and promising.

**Strengths:**

This work generalizes the single parameter bar codes to multi-parameter bar codes, introduces general vectorization techniques for signed barcodes, and prove Lipschitz-continuity results to ensure the robustness of the proposed pipeline. The theoretic results are important and convincing.
The paper is well written, the key concepts are explained thoroughly, the proofs of the main theorems are clear and in details. The experimental results are analyzed in details.

**Weaknesses:**

The motivation for multi-parameter persistent homology could be further emphasized. From theoretical point of view, it is unclear what extra-information can multi-parameter PH bring compared to single-parameter PH.

It will be helpful to add some experiments to validate the stability theorems, and estimate the Lipschitz-constant.

The definition of the Kantorovich-Rubinstein norm for point measures in proposition 1 is inconsistent with that in definition 3. The definition 3 allows point mass splitting, namely one source point can be mapped into multiple target points, hence \psi is a transportation scheme; but in proposition 1, each source point is mapped to a single target point, \gamma is a transportation map. This needs to be clarified.

**Questions:**

1. Whether single-parameter persistent homology is equivalent to multi-parameter persistent homology ? What extra-information can multi-parameter PH contributes?
2. The Kantorovich-Robinstein norm in Definition 3 is inconsistent with that in proposition 1, under what conditions are they equivalent in general ?
3. The generalization of theorem 1 to higher dimension is open, is the difficulty intrinsic or just technical ?

**Limitations:**

The authors have adequately addressed the limitations, which are helpful for practical applications.

---

> ### Author Rebuttal · Authors · 2023-08-09
>
> Thank you for your time and feedback.
>
> - *Motivation for multiparameter persistence and extra info compared to one-parameter.*
> Thank you very much for pointing out the lack of references motivating multiparameter persistence and its relationship to one-parameter persistence.
> To address this, we will add the following short paragraph right after line 47:
>
> >There are many applications of TDA where multiparameter persistence modules are more natural than, and lead to improved performance when compared to, one-parameter persistence modules.
> >These include
> >noisy point cloud data
> >[Vipond, Bull, Macklin, Tillmann, Pugh, Byrne, Harrington. Proc Natl Acad Sci USA 2021],
> >where one parameter accounts for the geometry of the data and another filters the data by density;
> >multifiltered graphs
> >[Demir, Coskunuzer, Segovia-Dominguez, Chen, Gel, Kiziltan. NeurIPS 2022],
> >where different parameters account for different filtering functions;
> >and time-varying data
> >[Chen, Segovia-Dominguez, Coskunuzer, Gel. ICLR 2022],
> >where an extra parameter accounts for time-dependence.
>
> - *Definition 3 vs Proposition 1.*
> Proposition 1 is stating a property of the Kantorovich-Rubinstein norm, namely that, when evaluated on a point measure (i.e., a measure given by an integer linear combination of Dirac deltas), there always exists an optimal coupling that does not split masses.
> We gave details in the originally submitted appendix, but we agree that the fact that this is a property and not a different definition should be further emphasized to avoid confusions.
> We will add the following sentence right before Proposition 1 with that goal:
>
> > The following result says that, for point measures, the computation of the Kantorovich--Rubinstein norm reduces to an assignment problem, and that, in the case of point measures on the real line, the optimal assignment has a particularly simple form.
>
> - *Theorem 1 for higher $n$.*
> This is a very interesting question.
> We know of an extension for arbitrary $n$ but with a Lipschitz constant that depends on the simplicial complex $K$; although not as strong as the cases $n = 1,2$, this result does provide justification for the method for higher parameters.
> However, the proof is quite technical and, unlike the cases $n=1,2$, it does not follow easily from previous work and requires several new definitions; for this reason we decided to omit it.
> An extension for arbitrary $n$ with a Lipschitz constant that is independent of the simplicial complex $K$ seems plausible and is the subject of current work.
>
> - *Experimental validation of stability.*
> Thank you for suggesting this.
> Please see the rebuttal PDF we have submitted where such an analysis is carried out for the pipeline (multifiltered complex) -> (Hilbert signed measure) -> (convolution vectorization), showing that our vectors are indeed stable and also provide a quite strong lower bound.
> If the paper is accepted, in the final version we will include an analysis for the three constructions: (multifiltered comples) -> (Hilbert signed measure), (Hilbert signed measure) -> (convolution vectorization), (Hilbert signed measure) -> (sliced Wasserstein kernel).

---

> ### Comment · Area_Chair_rsfV · 2023-08-18
>
> Dear reviewer,
>
> Please **briefly acknowledge the rebuttal** by the authors and consider updating your score—we want to avoid borderline scores for reviews, and the discussion phase will close soon. If you have any additional questions to the authors  please ask them **now**.
>
> Thanks,\
> Your AC

---

### Author Rebuttal · Authors · 2023-08-10

We sincerely thank all the reviewers for their time and feedback.
We have responded to their questions, and we will be happy to provide further clarifications, if required.

Only a couple of short paragraphs (included in the responses to specific reviewers, below) are required in the main body of the paper to address the reviewers' feedback.
With respect to the appendix, an experiment will be added following a suggestion by reviewer 61ca (a short version of the experiment is in Fig. 1 of the rebuttal PDF).

---

### Decision · Program_Chairs · 2023-09-21

**Decision:**

Accept (poster)

**Comment:**

Reviewers unanimously agreed on the relevance of the proposed work. Making multiparameter persistence amenable to more efficient calculations promises novel integrations into modern machine learning techniques and novel insights. With most reviewers already opting for acceptance of the paper, it was a pleasure to see an enthusiastic and in-depth rebuttal by the authors. While some concerns could not be alleviated, they mostly pertain to the overall question of the use of persistent homology in machine learning—answering those is beyond the scope of the paper, but the authors are encouraged to think about more ways to contextualise their work vis-à-vis the machine learning community at large.

I am thus very happy to **endorse this work for publication and presentation at the conference**. I encourage the authors to use the time to revise their paper in terms of accessibility. While the overall efforts and the initial state were already lauded by reviewers, I also agree with reviewer `csyn` that additional work concerning accessibility is encouraged since it makes this exciting topic available to a larger audience.

The authors are also encouraged to consider additional baselines, as mentioned by reviewer `VRYZ`.